# Spatially-distributed Acoustic Parameter Estimation

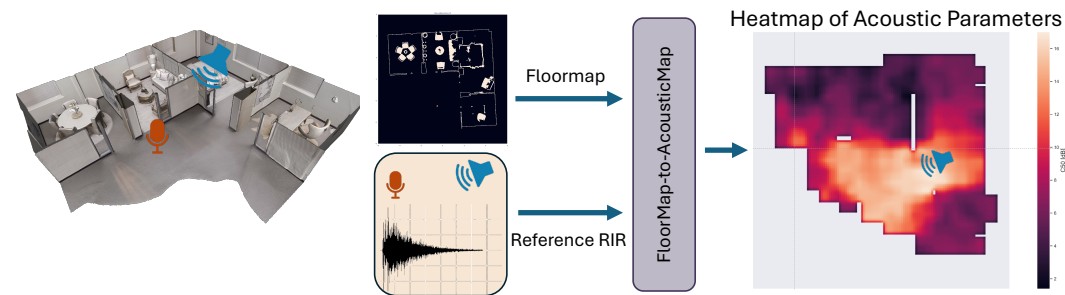

Figure 1: In this work, we study the problem of novel view acoustic parameter estimation. We predict 2D spatially distributed acoustic parameters for an unseen scene as an image-to-image translation task, using a simple floormap and a reference room impulse response as geometric and acoustic input respectively.

## Abstract

The task of Novel View Acoustic Synthesis (NVAS) – generating Room Impulse Responses (RIRs) for unseen source and receiver positions in a scene – has recently gained traction, especially given its relevance to Augmented Reality (AR) and Virtual Reality (VR) development. However, many of these efforts suffer from similar limitations: they infer RIRs in the time domain, which prove challenging to optimize; they focus on scenes with simple, single-room geometries; they infer only single-channel, directionally-independent acoustic characteristics; and they require inputs, such as 3D geometry meshes with material properties, that may be impractical to obtain for on-device applications. On the other hand, research suggests that sample-wise accuracy of RIRs is not required for perceptual plausibility in AR and VR. Standard acoustic parameters like Clarity Index (C50) or Reverberation Time (T60) have been shown to capably describe pertinent characteristics of the RIRs, especially late reverberation. To address these gaps, this paper introduces a new, intermediate task centered on estimating spatially distributed acoustic parameters, that can be then used to condition a simple reverberator to generate RIRs for arbitrary source and receiver positions. The approach is modeled as an image-to-image translation task, which translates 2D floormaps of a scene into 2D heatmaps of acoustic parameters. We introduce a new, large-scale dataset of 1000 scenes consisting of complex, multi-room apartment conditions, and show that our method outperforms statistical baselines significantly. Moreover, we show that the method also works for directionally-dependent (i.e. beamformed) parameter prediction. Finally, the proposed method operates on very limited information, requiring only a broad outline of the scene and a single RIR at inference time.

## 1 Introduction

As we work towards more augmented and virtual reality technologies for effective entertainment, communication, and telepresence, we find that achieving plausible acoustics is a critical requirement for immersion. Research has demonstrated that rendering virtual sounds with accurate models of the acoustics in an enclosed space - models of how sound moves through corridors, bounces off of walls and furniture to reach our ears at slightly different times – enables greater intelligibility, sound localization accuracy, and sense of co-presence and connection (Anuar et al., 2024; Vorländer, 2020; Pind et al., 2018; Isak de Villiers Bosman & Hamari, 2024; Broderick et al., 2018).

One way to achieve realistic acoustics is to infer *Room Impulse Responses* (RIRs) automatically in unseen environments. RIRs – defined as the acoustical transfer function between an emitter and receiver at arbitrary positions in the scene – can be used to fully describe the acoustical behavior of a physical space. Recently, this problem has gained traction in the machine learning community, under an emerging area of study known as *Novel View Acoustic Synthesis* (NVAS). First introduced by Chen et al. (2023), NVAS is defined as the task of inferring RIRs for unknown source and receiver positions within a room, or within new rooms entirely, using multimodal information that describes the geometry and material properties of the scene. While research in this area has been promising, previous work is limited by poor generalization to new scenes, poor handling of complex, real-world geometries, single-channel RIR estimation that ignores directional dependencies, and a requirement for high-dimensional input data, such as complete 3D meshes with labeled material properties.

We consider several novel directions to overcome these limitations. Firstly, we study more elaborate scene geometries that better reflect real-world spaces, namely multi-room apartments. Multi-room apartments are typically characterized by complex reverberation and sound transmission patterns, such as inhomogeneity and anisotropy (Götz et al., 2023; Götz et al., 2023), making them challenging to model. Secondly, we intuit that modeling complete RIRs are not necessary for perceptual plausibility, especially for multi-room apartments. Previous work suggests that acoustics parameters like Clarity Index (C50) and Reverberation Time (T60), can be viable intermediate prediction targets, used to inform downstream reverberators (Välimäki et al., 2016; Carpentier et al., 2013; Schlecht & Habets, 2017; Dal Santo et al., 2024) to generate plausible RIRs. Finally, we posit that conditioning a machine learning model with minimal acoustical context, like an RIR randomly sampled from the scene, can reduce the need for detailed geometry input that may be impractical to obtain in real-world scenarios.

In this work, we build upon previous research in NVAS to introduce and present an approach to a new task, *Spatially-distributed Acoustic Parameter Estimation* (SAPE). Unlike in NVAS, we do not predict RIRs directly; instead, we predict spatially-distributed acoustic parameters of scene. These inferred acoustic parameters can then be used to generate plausible RIRs with state-of-the-art algorithmic reverberators; we qualitatively demonstrate this process to illustrate a complete pipeline for acoustic rendering. To predict 2D acoustic parameters, we use limited geometric information and do not require precise material properties. We use a single, randomly chosen RIR as a calibration input to ground the model's understanding of the acoustic environment, and frame the task as an image-to-image translation problem. In this paper, our key contributions are the following: **(1)** We introduce a new task, SAPE, as an alternative to the previous NVAS task, which does not require detailed geometric information as input and is grounded in perceptual acoustics; **(2)** We construct a new dataset to study the task, called MRAS (Multi-Room Apartment Simulations); **(3)** We propose a deep learning model to solve the task, which consumes a 2D floormap and sample RIR to estimate an acoustic parameter map; **(4)** We demonstrate that our model outperforms baselines on the new task and achieves state-of-the-art benchmarks on existing tasks; and **(5)** We extend our model to address the previously unstudied case of spatially-dependent acoustic parameter prediction.

## 2 RELATED WORK

### 2.1 GEOMETRY-BASED ROOM IMPULSE RESPONSE SIMULATION

In general, RIRs can be obtained either by measuring real-life environments or approximating measurements with simulations. These simulations typically require detailed inputs, such as the full scene geometry (e.g., a 3D mesh) as well as the acoustical properties of each surface, including absorption and scattering coefficients. The simulations techniques can be divided in two categories, geometric acoustics and wave-based methods. Geometric acoustics are usually less accurate (especially for low frequencies) but less expensive, and include methods like image source and ray tracing (Allen & Berkley, 1979; Vorländer, 1989; Savioja & Svensson, 2015). Conversely, wave-based methods, like Finite Element Method (FEM) (Marburg & Nolte, 2008) and Finite-Difference Time-Domain (FDTD) (Botteldooren, Dick, 1995), provide higher accuracy at the expense of significantly higher computational costs. More recently, deep learning methods have been used to synthesize RIRs directly, either from the full 3D mesh with materials (Ratnarajah et al., 2022a), or from simplified geometry (Ratnarajah et al., 2021; 2022b). Additionally, some research has incorporated inductive biases to aid neural networks in understanding sound propagation, using techniques such as Physics-Informed Neural Networks (PINNs) (Karakonstantis et al., 2024; Borrel-Jensen et al., 2021; 2024; Miotello et al., 2024). Although these deep learning approaches can generate high-fidelity RIRs, they still require detailed knowledge of the full geometry and material properties, which can be challenging to obtain for many practical applications.

## 2.2 Novel View Acoustic Synthesis (NVAS)

Recently, several works that focus on predicting acoustics have been proposed. In general, the goal of these works is to synthesize RIRs for arbitrary source and receiver positions in a scene. The main differences among these works is the type of inputs, outputs, and conditioning. The inputs can include scene geometry, reverberant audio, RIR, and natural images of the scene, among others. The outputs can include single-channel RIRs or binaural RIRs, which can enable the rendering of reverberant audio via convolution with an anechoic signal such as speech.

**Multimodal Conditioning**    (Chen et al., 2023) introduced the NVAS task, utilizing reverberant binaural audio along with a video image of the scene to synthesize reverberant binaural audio at new positions and orientations within the scene. The multimodal approach adds conditioning that improves performance given the limited information available from a single binaural input. Further advancements are made in (Ahn et al., 2023), which performs joint source localization, sound separation, and dereverberation using a reconstructed 3D geometry of the scene and recordings from multiple sources and receivers. The authors first extract dry, separated sound sources and then synthesize reverberant audio at new positions by convolving them with simulated RIRs. A similar model in (Ratnarajah et al., 2023b) utilizes multimodal input that includes panoramic images, depth maps, and materials maps to estimate RIRs. However, these input modalities have limitations. First, there is little acoustic information contained in the input, unless materials are explicitly specified. More importantly, the task of mapping images to reverberation features is ill-conditioned at best, as similar-looking rooms can have vastly different acoustics.

The idea of using a set of measured RIRs to extract acoustic information has also been explored. Signal processing techniques have been used to interpolate RIRs from sparse measurements using a modal architecture to estimate room-mode parameters (Das et al., 2021). Closely related to our work, (Majumder et al., 2022) introduced Few-ShotRIR, a multimodal approach that uses a set of reference RIRs together with egocentric images of the scene to condition a transformer decoder, synthesizing binaural RIRs at novel source-receiver positions within the scene. Later work by (Wan et al., 2024) uses a set of measured RIRs to inform a differential path-tracing algorithm that learns the geometry of the scene, enabling the generation of RIRs for novel source-receiver positions.

**Acoustic Matching**    When the only available input is reverberant audio rather than an RIR, a process known as acoustic matching can be employed to implicitly extract the acoustic characteristics from the audio and then transfer them to another recording. This technique allows for the transformation of reverberant audio from one acoustic environment to another or from a specific source-receiver pair to a different one. Early work utilizing encoder-decoder architectures based on WaveNet networks was able to implicitly extract RIRs and render high-quality time-domain audio (Su et al., 2020; Koo et al., 2021). Subsequent research has refined these techniques using more advanced architectures, including vector quantization and transformer subnetworks (Lee et al., 2023). Additionally, acoustic matching can benefit from supplementary conditioning, such as images of the source and target environments, to further improve the quality of synthesized audio (Somayazulu et al., 2023; Ratnarajah et al., 2023a). While acoustic matching can synthesize very high-quality reverberant audio, it is primarily limited to replicating the acoustic characteristics of an existing environment and cannot infer these characteristics for entirely new or novel scenes.

**Implicit representations**    Implicit representations encode geometric or other data using neural networks that map input coordinates to their corresponding values, such as occupancy, color, or density. This allows for continuous and high-resolution reconstructions of complex shapes and scenes without the need for explicit mesh structures or discrete grid storage. In acoustics, implicit representations have been utilized for interpolation tasks by encoding acoustic properties into the neural network, enabling the rendering of Room Impulse Responses (RIRs) at novel source-receiver positions. However, these methods typically require a large number (thousands) of RIRs per scene and are primarily limited to interpolation tasks, with limited capabilities to generalize to completely novel scenes or complex geometries. For instance, Neural Acoustic Fields (NAF) (Luo et al., 2022), a NeRF-based implicit representation of an acoustic scene, needs to be trained individually for each scene and requires a very dense grid of source and receiver positions, thus only generalizing to unseen areas within the same scene. Further advancements, such as INRAS proposed by (Su et al., 2022), introduced dedicated modules to learn source and receiver positions as well as indirect room geometry based on a few bouncing points. Although INRAS improves upon the NAF approach, it still has limited generalization capabilities and does not perform well in multi-room scenes or scenarios where the receiver is occluded. Other work like DeepNeRAP (He et al., 2024) also introduces sound propagation physics to the model to reduce the need for large-scale RIR data, but this approach also has a limited ability to generalize to unseen geometries.

### 2.3 ROOM ACOUSTIC PARAMETER ESTIMATION

Acoustic parameters are standardized metrics that measure specific properties of RIRs, and describe key acoustic characteristics of indoor environments such as reverberation time, or speech intelligibility. While these parameters can be computed directly from RIRs, an interesting research area focuses on estimating them from other inputs when RIRs are not available. Reverberation time, for instance, can be roughly predicted from room geometry and absorption coefficients using simple formulas (Sabine, 1922; Neubauer, 2001; Eyring, 1930), or spherical maps of absorption with a machine learning model (Falcón Pérez, et al., 2019; 2021). Additionally, acoustic parameters can also be estimated from reverberant audio, usually speech, in a process known as blind estimation. Typical models for this task include transformers (Wang et al., 2024a;b), CRNNs (López et al., 2021), CNNs (Ick et al., 2023), and variational autoencoders (Götz et al., 2023; 2024). Finally, some studies explore related tasks that estimate geometrical properties of the room instead, such as room dimensions (Yuanxin & jeong, 2023) or room identification (Peters et al., 2012). Most previous work focuses on estimating acoustic parameters for a single source and receiver pair, that is at a single location within the scene. In contrast, our work aims to estimate the acoustic parameters distributed across the whole scene, for an unseen scene and arbitrary source position.

## 3 METHOD

### 3.1 TASK DEFINITION

In this paper, our ultimate goal is to predict the RIR corresponding to any pair of source and receiver locations, in any scene, given reference information about the scene geometry and acoustics. For a given 3D sound scene $D$, we denote the sound source (or *emitter*) locations as $E \in \mathbb{R}^2$, the receiver locations as $R \in \mathbb{R}^2$, and the corresponding impulse response as $h \in \mathbb{R}^{N \times T}$, where $N$ and $T$ are the number of channels and time steps, respectively. Formally, we are interested in solving the task:

$$\phi(E_r, R_r, h_{(E_r, R_r)}, E_t, R_t, D) \to \hat{h}_{(E_t, R_t)}, \tag{1}$$

where the subscripts $r$ and $t$ denote reference and target locations, and $D$ is comprised of all of the acoustic properties of a scene, including geometry and materials. However, in this work, we approach this task with a simplified, surrogate task. We attempt to estimate an acoustic parameter heatmap, corresponding to a target emitter location and all receiver locations, from a 2D floormap. We denote the acoustic map as $\mathcal{A}_{E_t} \in \mathbb{R}^{H \times W}$, and the floormap as $\mathcal{F} \in \mathbb{R}^{H \times W}$. Our task then becomes:

$$\phi'(\mathcal{F}, E_r, R_r, h_{(E_r, R_r)}, E_t) \to \mathcal{A}_{E_t}, \tag{2}$$

The inferred acoustic heatmaps can then be used to condition a reverberator, $m$, to render a plausible RIR, as:

$$m(\mathcal{A}_{E_t}, R_t) \to \hat{h}_{(E_t, R_t)}. \tag{3}$$

### 3.2 FEATURE AND LABELS EXTRACTION

**Floormaps and Reference RIRs** In this work, we consider floormaps as a lightweight source of information about the geometry of scene, which is consumed by our model. Floormaps can be obtained easily for real-world rooms from building construction plans or site layouts, or simple manual measurements. Other methods to extract floormaps without knowledge of the complete 3D geometry (Mura et al., 2021; Gueze et al., 2023; Majumder et al., 2023) have also been well-studied. However, since we use synthetic scene data in this work, we use the 3D meshes of the scenes to help generate the floormaps. To extract the floormaps we take the full 3D mesh of the scene, and create a 2D map by slicing at a specified height. Our goal is to capture scene boundaries and internal subdivisions, but avoid details that have little impact on the late reverberation, like furniture. In practice, we use a fixed slice height of about 0.5 or 1 meter below the ceiling of the scene. The selected slice is then digitized into a binary 2D map of size $128 \times 128$. To provide acoustic context in addition to the geometric context, we also provide a reference RIR, from an arbitrary source and receiver position, as input to the model. We encode the emitter and receiver position of this IR by marking their locations on another 2D binary map, concatenated as an additional channel to the floormap. Finally, we compute the magnitude spectrogram of this reference IR and

concatenate it as the final channel. We truncate the RIRs to 1 second, at 24 kHz and use 128 Mel bins with a hop size of 188 samples so that the spectrogram also becomes a $128 \times 128$ matrix. Therefore, all of the input features are represented as a single, multi-channel image.

**Acoustic Heatmaps**  We next construct acoustic heatmaps that represent the spatially distributed acoustic parameters in the scene, and serve as our labels for supervised learning. These parameters are computed from RIRs captured at a discrete set of receiver locations. To map the parameters from these sparse locations into a continuous and smooth heatmap, we use masked average pooling, where only the pixels that are active contribute to the pooling operation. Finally, to ensure a smooth response, we apply a 2D low pass filter via convolution with a Gaussian kernel (of size 9x9 and standard deviation of 1.) . An example of the masked average pooling operation is shown in Figure 2. Unlike a Voronoi map, this operation creates continuous maps without hard transitions. This process is repeated for each desired acoustic parameter and frequency band. The resultant maps are stacked along the channel dimension. Additional examples of these acosutic heatmaps are shown in Appendix B.

The acoustic parameters we use are well established. In this study, we focus on two main categories: 1) energy decay rates, and 2) early-to-late energy ratios. The former measures the time it takes for the energy of the RIR to decay a specific level. The latter measures the ratio of energy between the early reflections and the late reverberation of the RIR at a predefined transition point. We compute the parameters following the methods as defined in the standard (ISO 3382). For our experiments, we focus on EDT, $T_{30}$ for decay rates and DRR, $C_{50}$ for energy ratios, which can be considered statistically sufficient to describe human perception of late reverberation in indoor environments (Helmholz et al., 2022; Florian et al., 2023; Neidhardt et al., 2022). Nevertheless, our approach is flexible and could be applied to other acoustic parameters.

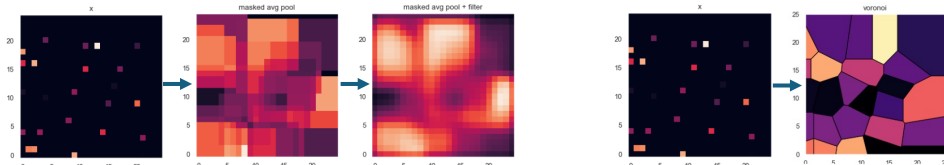

Figure 2: (left) Acoustic heatmap processing using a low passed masked average pooling operation. A set of acoustic parameter values at sparse receiver locations gets pooled, considering only locations with valid data. Then, a low pass filter is applied. We compare our labels to a Voronoi map (right) of the same set of receivers, which generates fragmented maps with hard transitions.

### 3.3 Floormaps to Acoustic Heatmaps

We approach this task as an image translation task. In computer vision, the image-to-image translation involves transforming an image from one domain to another while preserving its original content (e.g. translating photographs to hand drawn sketches) (Isola et al., 2017; Zhu et al., 2017; Pang et al., 2022; Wang et al., 2018). Here we learn a mapping from the floormap and reference IR features to acoustic heatmaps that show the spatial distribution of acoustic parameters in the scene. Figure 3 shows the overall proposed solution. The neural network consumes a 2D floormap and reference RIR from an arbitrary source-receiver position as input features, and outputs a stack of heatmaps. Each heatmap corresponds to one acoustic parameter (one of $C_{50}$, DRR, $T_{60}$, EDT) at one frequency band (of 125, 250, 500, 1k, 2k, 4k Hz).

More formally, we construct and train a model via:

$$\underset{\phi'}{\arg\min} \sum_{d \in D} \mathcal{L}\big(\phi'(\mathcal{F}, E_r, R_r, h_{(E_r, R_r)}, E_t), \mathcal{A}_{E_t}\big) \tag{4}$$

where the loss is the pixel-wise mean absolute error between the target acoustic heatmaps and the predicted maps $\hat{\mathcal{A}}_{E_t}$, computed across $P$ acoustic parameters and $F$ frequency bands, defined as:

$$\mathcal{L} := \sum_{p \in P} \sum_{f \in F} \big|(\mathcal{A}_{E_t} - \hat{\mathcal{A}}_{E_t}) \odot M\big|, \tag{5}$$

where $M$ is a binary mask where pixels corresponding to valid (i.e., within the scene, with valid acoustic data, etc) values in the target heatmap $\mathcal{A}_{E_t}$ are set to 1, and 0 otherwise. The element-wise multiplication ($\odot$) ensures that

Figure 3: **Estimation of spatially distributed acoustic parameters as an image translation task**. First, 2D floormaps are extracted from the 3D model of the scene, consisting of a 2D vertical slice and a mask delimiting the total scene area. This geometric feature has no information about materials or acoustic properties. An RIR for an arbitrary source and receiver provides acoustic context for the scene, where the source position also acts as the target source. The features are then fed to a deep neural network to predict acoustic heatmaps for $p$ parameters at $f$ frequency bands.

the loss is computed only for valid pixels. The model $\phi'$ is a typical U-Net neural network based on ResNet block. The architecture is shown in detail in Appendix C.

## 4 DATASETS

We consider several public datasets that include both scene geometry and RIRs, including (Götz et al., 2021; Tang et al., 2022; Prawda et al., 2022). However, for our task, we require a dataset with a large number of unique scene geometries, with multiple sources per scene, a dense grid of receivers, and multi-channel responses. In addition, we require diverse scene geometries and acoustics throughout the dataset. We have a particular interest in multi-room scenes that represent typical indoor apartments, which are known for complex late reverberation phenomena (McKenzie et al., 2021; Billon et al., 2006). Therefore, we conduct experiments using one existing state-of-the-art dataset, SoundSpaces (Chen et al., 2020), and construct a novel dataset, the Multi-room Apartments Simulation (MRAS) dataset.

### 4.1 SOUNDSPACES

We first use the Soundspaces 1.0 dataset (Chen et al., 2020) with the Replica (Straub et al., 2019) scenes, in order to enable comparisons with previous work. The Replica set has 18 scenes in total, including 3 multi-room apartments, a studio apartment with 6 different furniture configurations, and other small, single-room, shoe-box scenes. The meshes are detailed, including all furniture and small objects inside. Rooms in the dataset contain a dense grid of source and receivers, with a uniform spacing of 0.3 m. Scenes typically contain approximately 250 sources and receivers, for a total of $250^2 = 60,000$ unique RIRs per scene. However, acoustic diversity is limited; for example, most of the scenes have $T_{30}$ value of around 0.6 seconds, with little variance across and within scenes. For further details regarding the statistics and types of geometries associated with this dataset, refer to Appendix A.

### 4.2 MULTI-ROOM APARTMENTS SIMULATION (MRAS)

The Multi-Room Apartments Simulation (MRAS) dataset is a novel multi-modal dataset created specifically for the task of estimating spatially-distributed acoustic parameters in complex scenes. It includes a large collection of

scene geometries, with dozens of unique source positions, and a dense grid of receivers. The scene geometries are generated algorithmically by connecting shoe-box rooms using two distinct patterns, and the RIRs are generated using the same acoustic ray-tracing methods as in (Chen et al., 2020). A key contribution of this work is the release of the MRAS dataset for public use. This includes the 3D meshes and the raw RIRs, as well as the pre-processed floormaps, acoustic parameters, and acoustic maps used in the experiments [1].

**Scene Generation**    We generate a total of 1000 scenes: 100 unique geometries using a linear pattern, 100 unique geometries using a grid pattern; where each geometry type is further configured with 5 different sets of materials. This results in $100 \times 2 \times 5 = 1000$ distinct acoustic scenes in total. The line pattern is built by connecting shoe-box rooms along a common boundary creating coupled-room scenarios. The grid pattern subdivides a large rectangular area into multiple connected rooms. In both cases, individual shoe-box rooms may have varying heights. Materials are randomly assigned to the floor, ceiling, and walls of each room. The materials are uniformly sampled from a set of realistic materials (e.g. carpet, concrete), plus two additional materials: a highly absorptive material (absorption coefficient $> 0.9$ for all frequency bands), and a highly reflective material (absorption coefficient $< 0.1$ for all frequency bands). Lastly, the doorframes that connect the rooms in a scene have random width, from a minimum of 0.9 meters up to as wide the wall it is located in. This creates scenes that have wide hallways instead of rooms connected via small openings. Although the scenes are constructed algorithmically, the geometries offer high acoustical complexity.

**Acoustic simulation**    The dataset has approximately 4 million RIRs, divided across 1000 scenes. For each scene, we sample 3 receiver positions per room to act as source positions. This sampling is done uniformly, without regard to the size of the room. We construct a dense grid of receivers with 0.3 m of spacing; positions less than 0.5 m from any boundary are removed. The RIR simulations are rendered in a 2nd-order ambisonics format. Further details regarding the distribution of acoustic parameters and sample geometries can be found in Appendix A.

## 5    EXPERIMENTS

### 5.1    BASELINES

We compare the performance of our model to some algorithmic baselines. The baselines are ordered in increasing order of oracle information available to compute the result. These include:

- **Average RIR** (AVG RIR): We randomly sample 500 RIRs from the dataset, compute the samplewise mean (i.d. time domain average) to create an overall representative RIR. Then we compute the acoustic parameters (per frequency band) on this representative RIR. Finally, we apply the values of the acoustic parameters to all pixels of all scenes uniformly. As a result, all of the scenes have the same values.

- **Average RIR, Same Scene** (SCENE AVG RIR): We follow the same process as the Average RIR baseline, but the sampling is done per scene; therefore each scene has its own representative RIR.

- **Input RIR** (INPUT RIR): We take the same RIR as the proposed model. Here, for each source we compute the acoustic parameters for one receiver, and apply them uniformly across the scene. We repeat this process for all receivers. This baseline can be considered a fair baseline, as it has access to the same acoustic information as the proposed model.

- **Random Acoustic Map, Same Scene** (SCENE RAND MAP): We sample a random source for the scene, and compute the pixelwise metrics with this.

- **Average Acoustic Map, Same Scene** (SCENE AVG MAP): We sample up to 100 sources for each scene, and compute the pixelwise mean. This is the strongest baseline as it utilizes information about all sources and receivers as input.

### 5.2    PERFORMANCE EVALUATION

To evaluate the performance, we report performance on individual acoustic parameters and additional metrics. For $C_{50}$ and DRR we measure the absolute error in dB. For EDT and $T_{30}$ we measure the proportional error. These metrics are computed pixel-wise, only for pixels where valid measurement data exists. However, these metrics

---

[1]The dataset will be published contingent on acceptance.

do not consider any spatial structure across the map. Therefore, we also include the Structural Similarity Index (SSIM) (Wang et al., 2004) to measure this. This metric is widely used in computer vision tasks, and offers some insights into our task. We report the mean and standard deviation of all error metrics.

Finally, we consider just noticeable difference (JND) metrics of 1 dB for energy ratio-based parameters ($C_{50}$ and DRR), and $10\%$ for decay time metrics ($T_{30}$ and EDT) (Bradley et al., 1999; Werner & Liebetrau, 2014). Although true JNDs are difficult to define (Florian et al., 2023), and depend on several factors (e.g. frequency band, stimulus type, sound level, directivity, scene acoustics) they provide a useful sanity check on our results.

## 5.3 EXPERIMENTAL SETUP

In our setup, we split the datasets into train/test partitions by scenes, where each scene is only available in either train or test. Furthermore, for Replica we manually select the scenes for each split, in order to keep a balanced distribution of scales and geometries. We train the model with a batch size of 128, minimizing the L1 loss (equation 5), using the ranger optimizer (Wright & Demeure, 2021) with a learning rate of 1e-3. All acoustic parameters are normalized such that about 90% of the values fall in the (-1, 1) range. We train the model until the validation loss stops decreasing for 3 consecutive validation steps. As data augmentation, we use random centered rotations and translations of the floormaps, constrained such that the whole scene is always visible in the floormap.

## 5.4 RESULTS

Table 1 shows the performance of the proposed model compared to the baselines on both datasets, for the case where the target contains 4 acoustic parameters ($C_{50}$, DRR, $T_{30}$ and EDT) at 6 frequency bands. First, we notice that the performance of the baselines mostly follows the amount of information available to them, where RIR-based baselines perform the worst. Secondly, the proposed model outperforms all baselines, although there are nuances. For parameters based on energy ratio ($C_{50}$, DRR) the proposed model achieves between 0.5 and 1 dB less mean error than the best baseline. However, for reverberation time metrics ($T_{30}$, EDT) the model is slightly worse than the best baseline, but still significantly better than the INPUT RIR baseline (which has the same acoustic information as the proposed model). This is because for any given scene, the reverberation time does not change significantly with source position. Therefore, the average of multiple acoustic maps of the same scene approximates the reverberation time quite well.

Furthermore, the overall trends are consistent across both datasets with two notable differences. Firstly, the performance for $C_{50}$ and DRR prediction on the RIR-based baselines is significantly better on the MRAS dataset as compared to the Replica dataset. This can be attributed to the complex multi-room geometries in MRAS, which include scenes with sparsely connected rooms. These geometries can result in cases where the direct path between the source and receiver is very long, leading to low-energy IRs dominated by reverberation. Secondly, for $T_{30}$ prediction, the baselines are higher-performing on Replica than on MRAS. This is because the MRAS dataset exhibits much higher acoustic variance, causing $T_{30}$ to cover a wider range. Despite these differences, the proposed model performs consistently across both datasets.

A qualitative example is shown in Figure 4, and detailed model ablations are discussed in Appendix E. The ground truth $C_{50}$ and DRR show strong dependency on the proximity to the source, while $T_{30}$ is much more uniform across the scene. The baseline fails to capture the spatial variance, producing uniform values instead. In contrast, the proposed model successfully captures patterns such as line of sight and proximity to source. However, the output appears to be smoother and fails to reproduce fine-grained spatial variations.

## 5.5 SPATIALLY-DEPENDENT ACOUSTIC PARAMETERS

The proposed method is flexible and can be adapted to different types of acoustic parameters. An interesting case is the use of spatially-dependent acoustic parameters, that overall can provide a more complete characterization of a scene by also considering the direction of sound (Campos et al., 2021; Meyer-Kahlen & Schlecht, 2022; Prinn et al., 2025). As an illustrative task, we predict a single parameter, $C_{50}$, at fixed directions from each location in the scene. To do this, we first take the 2nd-order ambisonic RIRs and beamform to 5 fixed orientations in the scene (azimuth only), about $72 \deg$ apart from each other. Details of the beamforming process and ground truth examples are shown in Appendix D.

Table 1: Experimental results for the prediction of 4 omnidirectional acoustic parameters aggregated over 6 frequency bands, given by the mean and standard deviation.

| Model | Dataset | Fold | $C_{50}$ (dB) ↓ | $T_{30}$ (%) ↓ | DRR(dB) ↓ | EDT(%) ↓ | loss ↓ | SSIM ↑ |
|---|---|---|---|---|---|---|---|---|
| | | | | | Metrics | | | |
| AVG RIR | Replica | 1,2,4 | 11.48± 5.62 | 33.57±13.96 | 7.30± 5.63 | 69.31±26.81 | 0.71± 0.61 | 0.14± 0.08 |
| SCENE AVG RIR | Replica | 1,2,4 | 9.80± 4.34 | 27.03±10.18 | 6.82± 5.05 | 69.39±26.72 | 0.66± 0.60 | 0.14± 0.09 |
| INPUT RIR | Replica | 1,2,4 | 3.82± 2.38 | 16.91± 7.75 | 2.61± 1.37 | 38.14±20.09 | 0.22± 0.18 | 0.30± 0.12 |
| SCENE RANDOM MAP | Replica | 1,2,4 | 3.51± 1.48 | 8.10± 5.60 | 2.37± 0.94 | 21.19±12.25 | 0.15± 0.10 | 0.40± 0.10 |
| SCENE AVG MAP | Replica | 1,2,4 | 2.59± 0.93 | **5.86 ± 4.21** | 1.71± 0.63 | 15.25±14.99 | **0.10 ± 0.07** | **0.54 ± 0.07** |
| Ours | Replica | 1,2,4 | **1.73 ± 0.85** | 10.77± 5.85 | **1.37 ± 0.48** | 17.52±57.98 | 0.10 ± 0.06 | 0.50± 0.08 |
| AVG RIR | MRAS | 1 | 3.86± 1.72 | 59.22±52.88 | 2.11± 0.83 | 40.35±26.16 | 0.22± 0.14 | 0.46± 0.09 |
| SCENE AVG RIR | MRAS | 1 | 3.44± 1.65 | 23.42±19.08 | 2.21± 1.03 | 32.72±20.60 | 0.17± 0.10 | 0.47± 0.08 |
| INPUT RIR | MRAS | 1 | 3.50± 2.22 | 25.91±19.24 | 2.42± 1.26 | 43.11±36.14 | 0.19± 0.13 | 0.46± 0.10 |
| SCENE RANDOM MAP | MRAS | 1 | 2.23± 0.98 | **11.50± 8.62** | 1.40± 0.53 | 20.42±13.54 | **0.10 ± 0.06** | **0.65 ± 0.09** |
| SCENE AVG MAP | MRAS | 1 | 2.93± 1.77 | 14.46± 1.42 | 1.88± 0.94 | 25.76±24.62 | 0.13± 0.10 | 0.54± 0.15 |
| Ours | MRAS | 1 | **1.87 ± 0.90** | 16.59± 9.95 | **1.33 ± 0.44** | 19.34±11.61 | 0.10± 0.06 | 0.58± 0.08 |

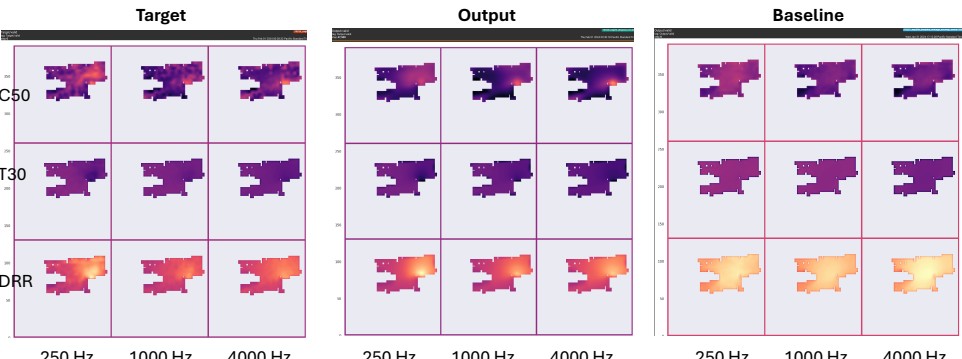

Figure 4: Example results from the SAPE task. The acoustic heatmaps produced by the proposed model reflect small details such as line of sight and distance to the source, while the SCENE AVG MAP baseline is mostly a uniform distribution across the scene.

Table 2 shows the results for the spatially dependent case. These show the same trends as in Table 1 but with overall worse performance (given the more difficult task), and with the heatmap-based baselines being noticeably better than the RIR-based baselines. Nevertheless, the proposed model performs similar to the main results, outperfoming all baselines. A key modeling difference is the inclusion of the pose channel. Since we train the model using random rotations of the scenes, the model must know the canonical orientation of the scene, to determine the rotated orientation of the fixed directions used when beamforming the ground truth. To address this, we add an additional channel to the input features, represented as a single line, which rotates in accordance with the floormap [2].

Table 2: Experimental results for the directional (spatially-dependent) case. The metrics show the mean and standard deviation of the prediction error of a single acoustic parameter, aggregated across 3 frequency bands, and 5 fixed orientations.

| Model | Dataset | Input | $C_{50}$ (dB) ↓ | loss ↓ | SSIM ↑ |
|---|---|---|---|---|---|
| AVG RIR | Replica | 2nd-order | 15.39± 5.17 | 0.75± 0.26 | 0.34± 0.06 |
| SCENE AVG RIR | Replica | 2nd-order | 10.84± 3.01 | 0.53± 0.15 | 0.39± 0.05 |
| INPUT RIR | Replica | 2nd-order | 4.87± 2.39 | 0.24± 0.12 | 0.48± 0.07 |
| SCENE RANDOM MAP | Replica | 2nd-order | 4.11± 1.57 | 0.21± 0.08 | 0.46± 0.11 |
| SCENE AVG MAP | Replica | 2nd-order | 3.09± 0.89 | 0.14± 0.04 | 0.63± 0.08 |
| Ours | Replica | 2nd-order | 3.12± 1.16 | 0.16± 0.06 | 0.57± 0.07 |
| Ours+pose | Replica | 2nd-order | **1.94± 0.92** | **0.10± 0.05** | **0.67± 0.10** |

---

[2] The 2nd-order RIR is also rotated to match the augmented rotation, as described in (Politis et al., 2012; Mazzon et al., 2019)

## 5.6 SINGLE-ROOM INTERPOLATION

Finally, to benchmark our approach on existing tasks, we compare our proposed model to state-of-the-art approaches for within-scene generalization. This is a different task than the SAPE task introduced in this paper; it is an interpolation task where the goal is to estimate acoustic parameters for novel source and receiver positions within the same scene. We compare to INRAS (Su et al., 2022) and NAF (Luo et al., 2022) (as reported). A key difference is that we explicitly estimate frequency dependent acoustic parameters, while both INRAS and NAF are implicit representations that output full RIRs, which are used to compute broadband acoustic parameters. To compare, we train our model with the same 3 scenes as mentioned in (Su et al., 2022), and include both their single-scene and multi-scene results. The results are shown in Table 3. Overall, the proposed model is significantly better than both INRAS and NAF. More interestingly, the proposed model achieves similar performance to INRAS even when using only 30 % of the data for training.

Table 3: Experimental results for the within-scene generalization case. All models are trained and tested with the same 3 scenes of the Replica dataset, where train/test splits have different receivers. Multi-scene models were trained on multiple scenes simultaneously, while others show the average across single scene models. We compute our baselines on this task for additional context.

| Model | Multi-scene | $C_{50}$ (dB) $\downarrow$ | $T_{30}$ (%) $\downarrow$ |
|---|---|---|---|
| AVG RIR | ✓ | $14.47 \pm 6.77$ | $40.61 \pm 13.58$ |
| SCENE AVG RIR | ✓ | $9.43 \pm 3.89$ | $24.74 \pm 9.52$ |
| INPUT RIR | ✓ | $3.89 \pm 1.94$ | $17.53 \pm 8.17$ |
| SCENE RANDOM MAP | ✓ | $3.62 \pm 1.54$ | $8.93 \pm 6.02$ |
| SCENE AVG MAP | ✓ | $7.95 \pm 2.26$ | $102.83 \pm 12.40$ |
| NAF (Luo et al., 2022) | ✗ | 1.05 | 3.01 |
| INRAS (Su et al., 2022) | ✗ | 0.6 | 3.14 |
| INRAS (Su et al., 2022) | ✓ | 0.68 | 4.09 |
| Ours | ✓ | $\mathbf{0.08 \pm 0.03}$ | $\mathbf{0.37 \pm 0.21}$ |
| Ours+split30/70 | ✓ | $0.55 \pm 0.19$ | $2.66 \pm 1.46$ |

## 6 LIMITATIONS

Our model may be improved by addressing noise within the dataset and the label and input generation process. For example, RIRs may contain errors due to failed simulations or ray propagation, or simulations that are physically inaccurate at low frequencies. Also, some metrics like DRR or C50 are not well defined for cases with no line of sight between source and receiver [3]. Additionally, our floormap extraction process may miss some properties, such as large pieces of furniture, that may impact the acoustics of the scene.

With regards to the output produced by our model, we note significant improvement over baselines and state-of-the-art approaches, but observe that fine-grained spatial variations are not perfectly captured. Further perceptual validation can be conducted to verify whether the output is sufficient for plausible auralization with state-of-the-art reverberators. Lastly, we expect to extend this work to real-world measurement data. While it is time-consuming to acquire this data at a large scale, we intuit that modeling a combination of simulated and measured data may lead to better perceptual outcomes.

## 7 CONCLUSION

In this paper we identify a new task, SAPE, that entails predicting acoustic parameters in unseen scenes and at arbitrary source and emitter positions, which can be used to condition reverberators to generate RIRs. We present a model architecture that is able to jointly estimate multiple spatially distributed acoustic parameters for multiple frequency bands, using limited geometric information in the form of a simple 2D floorplan, along with a reference RIR as input. We validate our approach on the SoundSpaces dataset, and a novel large scale dataset, MRAS simulating multi-room apartments.

---

[3]Some previous efforts ignore the direct sound and compute the time window based on the energy onset (Lee et al., 2024), but we choose to estimate the direct sound occurrence on the largest change of energy in the RIR.

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

## A   DATASET STATISTICS

Some examples of the geometries in MRAS are shown in Figure 5. The statistics of some relevant acoustic properties are shown in Figure 6. The areas and perimeter are approximated by counting the number of occupied pixels in the extracted floor maps and floor masks respectively (see Section 3.2). Therefore these values are the upper bounds of the true area and perimeter. In particular, the perimeter calculation includes internal subdivisions (e.g. room boundaries).

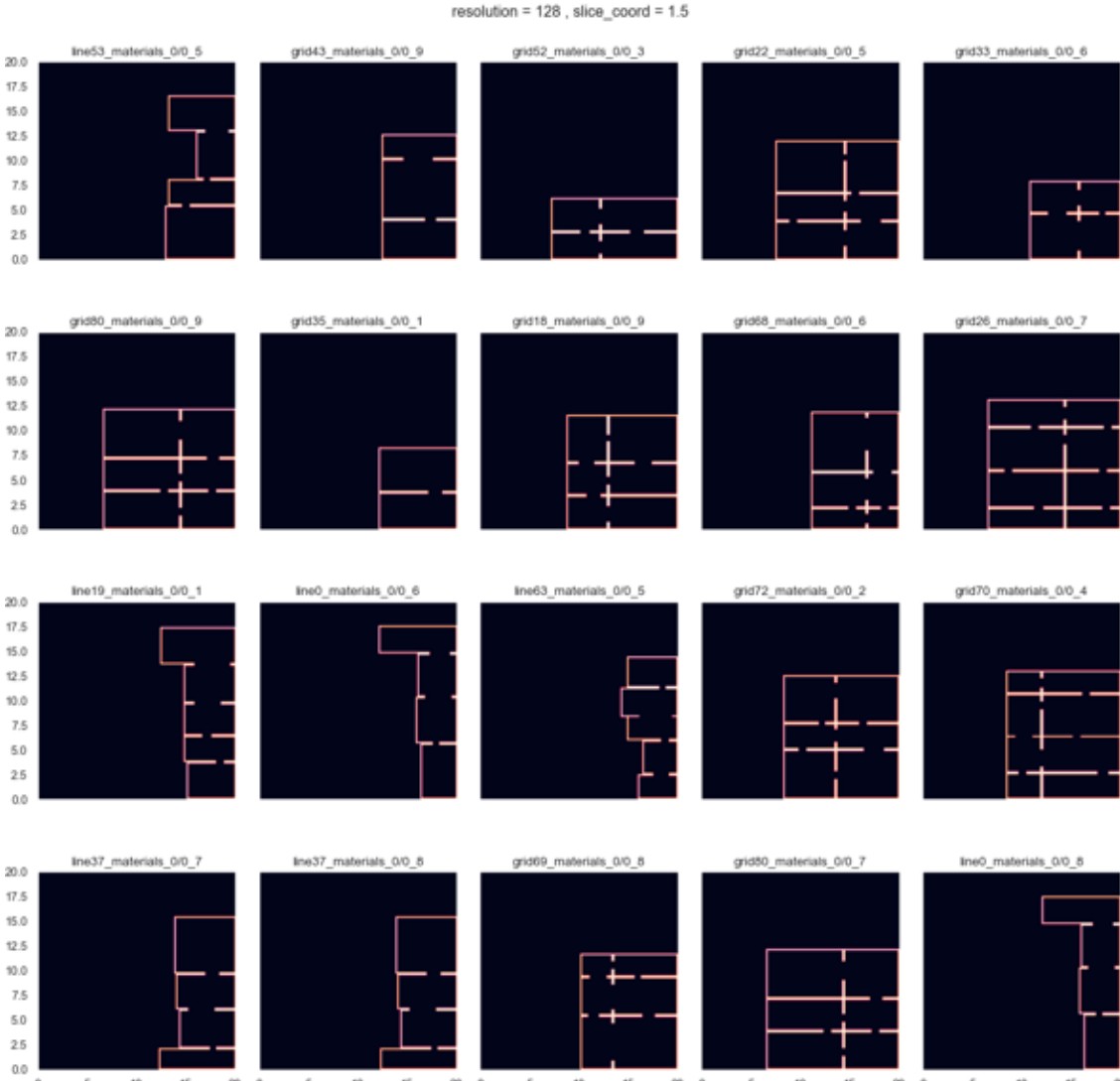

Figure 5: Example of the geometries of scenes from the MRAS dataset. The scenes simulate multi-room apartments by combining multiple shoe-boxes along different dimensions. The combinations are done in two patterns: 1) line, where the shoe-boxes are combined along a single axis; 2) grid, where a large rectangular area is subdivided into connected rooms. The doorframes connecting rooms can vary in width.

Figures 7, 8 show the joint and marginal distribution of acoustic parameters for the MRAS dataset. Compared to Replica (Figures 9, 10), there is a much larger range of $T_{30}$ values, as well as larger variance across frequency bands for $C_{50}$. This shows that MRAS presents much harder acoustical conditions.

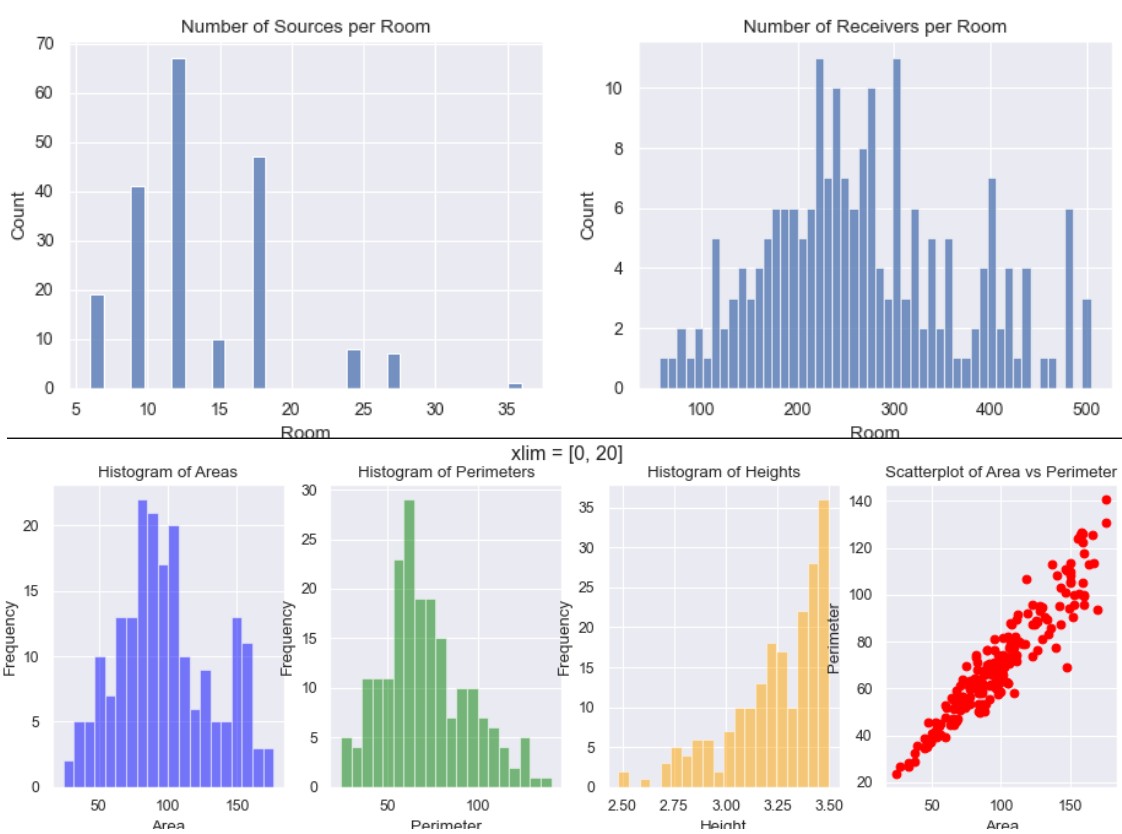

Figure 6: Statistics summarizing key geometrical properties of the MRAS dataset.

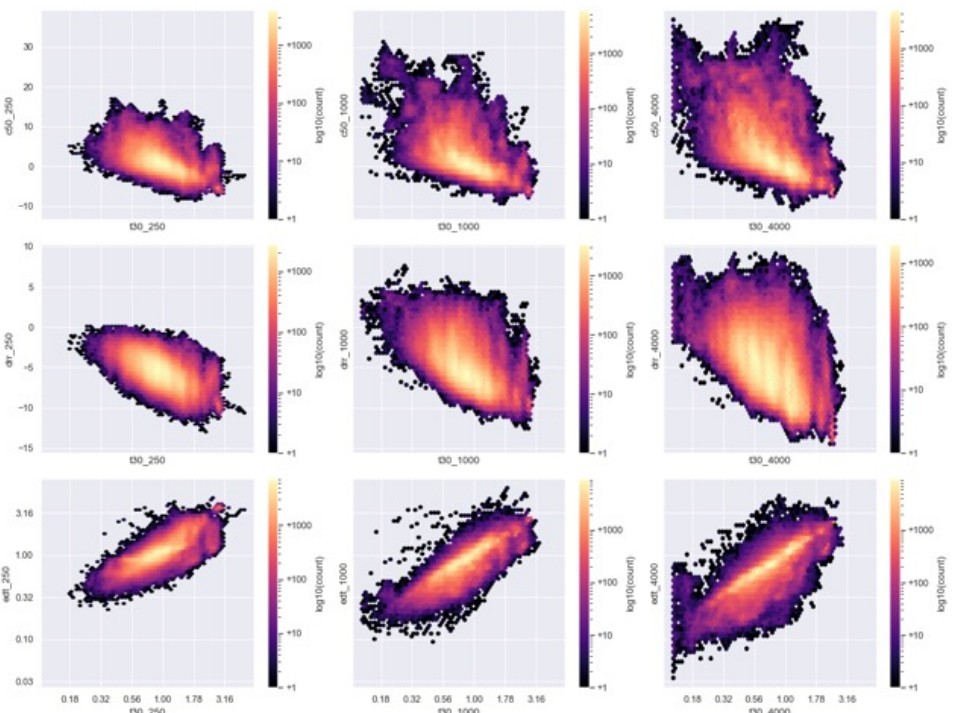

Figure 7: Joint distribution of acoustic parameters for the MRAS dataset for a random selection of 300 sources across all scenes. The dataset has a large acoustic variance. For example, $T_{30}$ covers a range from a few 100 ms (very dry) to over 3 seconds (highly reverberant).



Figure 8: Marginal distribution of acoustic parameters for the MRAS dataset, for a random selection of 300 sources across all scenes.

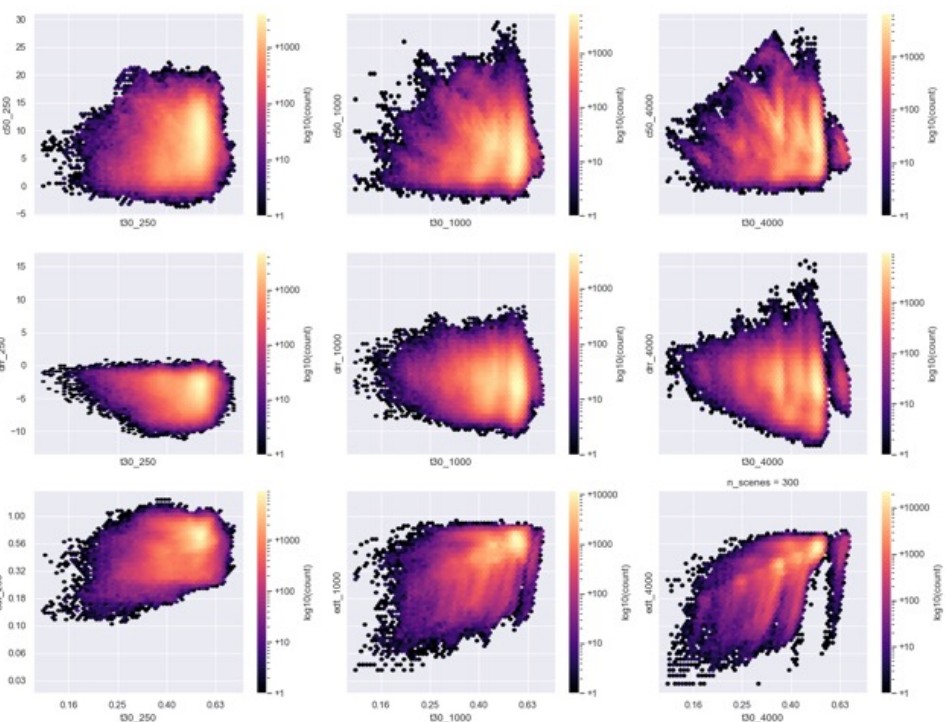

Figure 9: Joint distribution of acoustic parameters for the Soundspaces-Replica dataset for a random selection of 300 sources across all scenes, excluding *apartment_0*. Acoustic variety is limited. For example, the the majority of the scenes have $T_{30}$ around 0.6 seconds.

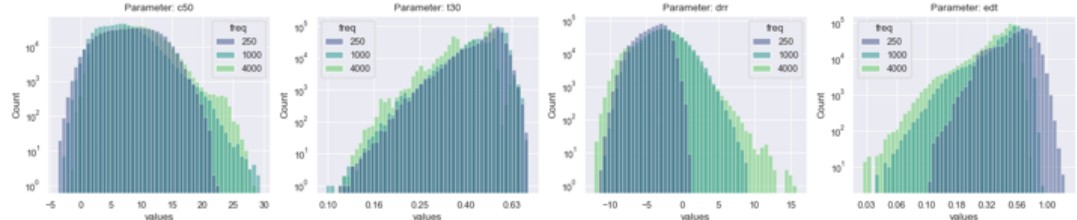

Figure 10: Marginal distribution of acoustic parameters for the Replica dataset, for a random selection of 300 sources across all scenes, excluding *apartment_0*.

## B  ADDITIONAL TARGET AND OUTPUT EXAMPLES

In Figure 11, and Figure 12, we give additional examples of target acoustic maps, model predictions, and input floormaps, corresponding to the experiment in Section 5.

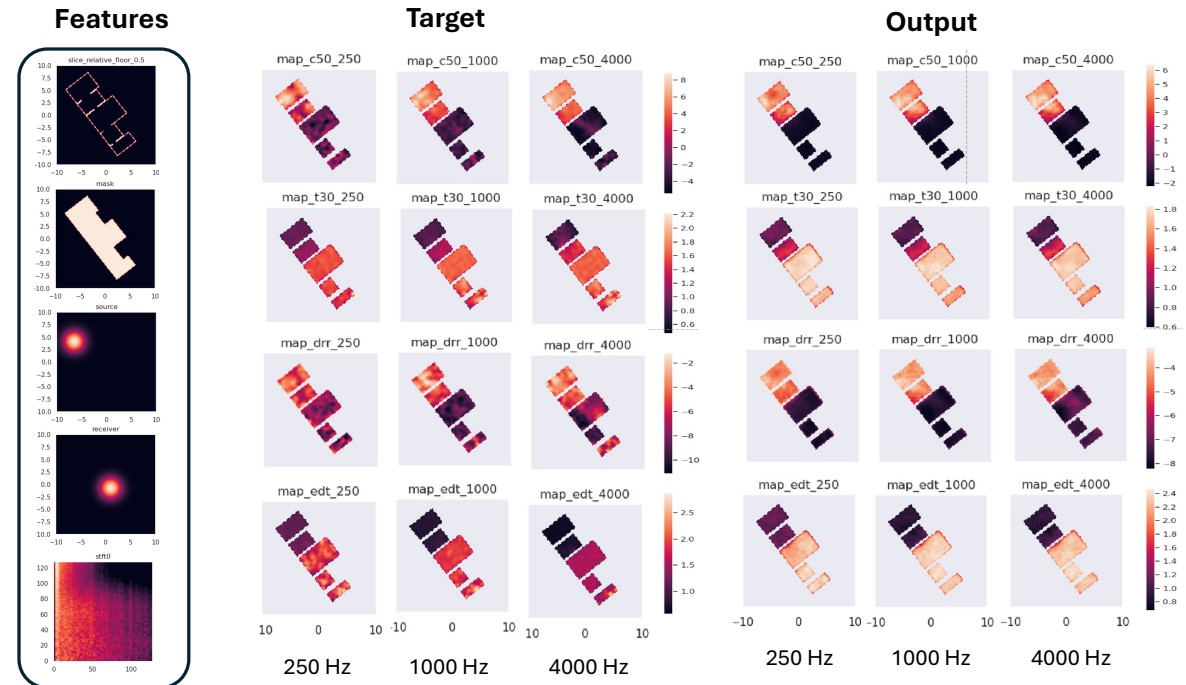

Figure 11: Example of a data point with features and target acoustic heatmaps from the MRAS dataset for a scene using the linear pattern. The variations in $T_{30}$ per room are clearly visible.

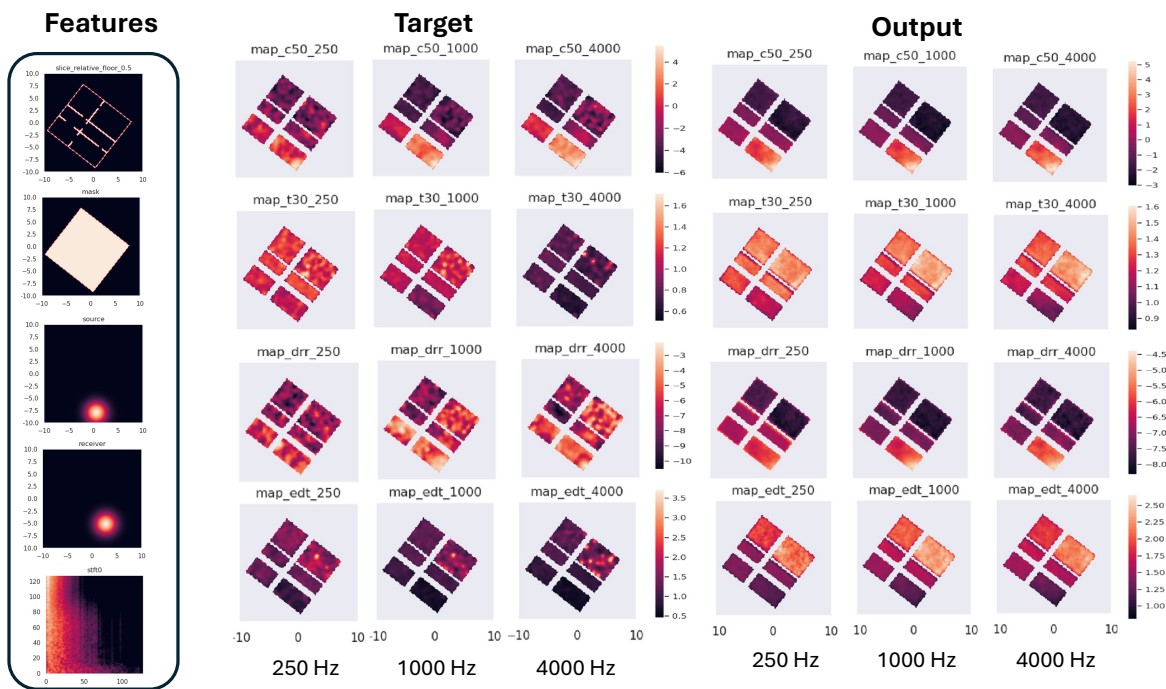

Figure 12: Example of a data point with features and target acoustic heatmaps from the MRAS dataset for a scene using the grid pattern. The rooms at the top of the scene are fairly isolated from the rest of the scene, creating challenging scenarios.

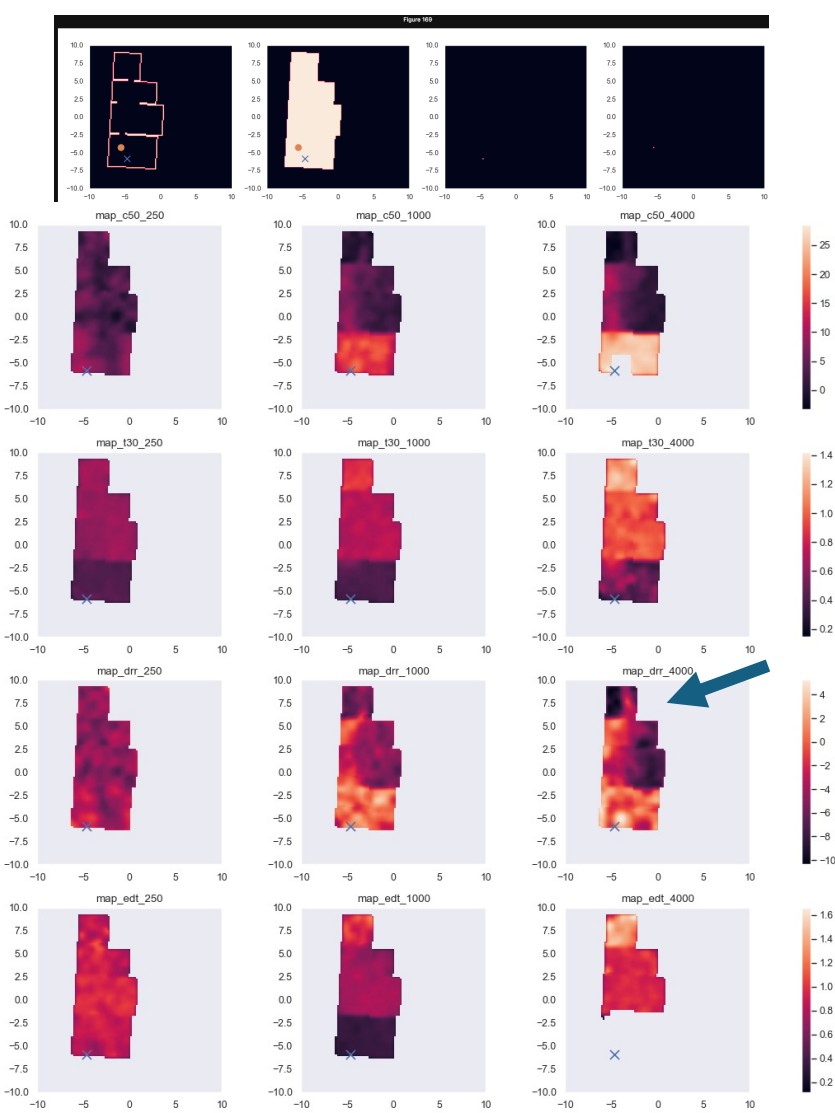

Figure 13: Example of a data point with features and target acoustic heatmaps from the MRAS dataset. This scene shows some interesting patterns. 1) The variation in $T_{30}$ per room are clearly visible, especially at high frequencies. 2) For DRR and $C_{50}$ the connections between rooms create line of sight patters towards the source. 3) (Blue arrow) DRR at high frequencies have high values even in the room farther away from the source, due to line of sight. 4) Overall, low frequencies are much more uniform for all acoustic parameters.

## C   NETWORK ARCHITECTURE

The architecture of our model is shown in Figure 14. This is a U-Net model consisting of symmetrical encoder and decoder networks built using typical ResNet blocks with skip connections between corresponding layers. Each block is comprised of [CONV2D, BATCHNORM, RELU, CONV2D, BATCHNORM] layers, along with a residual path via a 1x1 convolution. We replace the first CONV2D layer of each block with a COORDCONV2D layer, which adds two channels of positional encodings shown to improve generalization for tasks involving 2D coordinates (Liu et al., 2018). After each ResNet block, a resampling block consisting of a strided convolution and RELU non-linearity follows. The output of the encoder is fed directly into the decoder with no further processing. Our U-Net architecture is similar to those used in common image processing tasks like image-to-image translation (Isola et al., 2017), segmentation (Ronneberger et al., 2015), inpainting (Nazeri et al., 2019), and enhancement (Chen et al., 2018); as well as audio tasks such as source separation and speech enhancement (Kadandale et al., 2020; Slizovskaia et al., 2021).

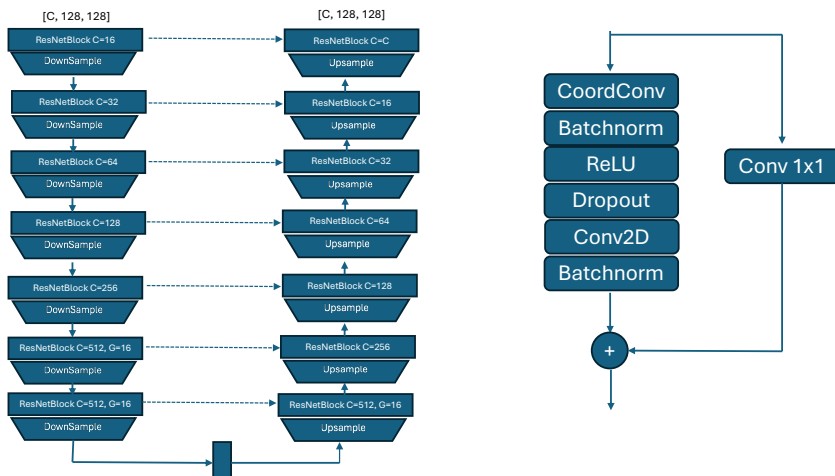

Figure 14: (left) Architecture of the U-Net and (right) ResNet block. $C$ is the number of output channels and $G$ is the number of groups.

## D   SPATIALLY-DEPENDENT PROCESSING (BEAMFORMING)

Beamforming in the ambisonic domain can be done in multiple ways (Zotter & Frank, 2019); here we use the Max-re beamformer weights, that maximize the front-back ratio (although there is no exact solution, the approximation is close enough for most perceptual analysis) (Zotter & Frank, 2012). In practice, this means that the side-lobes are wider than what is possible with other weighting techniques, but the back-lobes are minimized. The directional targets are computed by beamforming to 5 fixed directions in the scene, and then computing the acoustic parameters on the beamformed RIR. These fixed directions are static and do not change across scenes.

We alter the baselines slightly to account for this spatial task. For the RIR-based baselines, we first compute multi-channel sample-wise mean signals of the sampled RIRs. Then we apply the beamforming on these mean signals to compute the acoustic parameters. For the heatmap-based baselines, there is no beamforming needed as these baselines already sample the ground truth maps. We use only the Replica dataset due to its smaller size, given that the extraction of acoustic parameters for beamformed signals is expensive. An example of the spatial-dependent acoustic heatmaps is shown in Figure 16.

In addition, the input features are modified as mentioned in 5.5. An extra channel is added that denotes the canonical orientation of the scene. This channel is a single 2D line that points upwards originally. During training, the scene is randomly rotated, and this rotation is applied to this pose feature.

Finally, an example and comparison of the proposed model and the best performing baseline (SCENE AVG MAP) are shown in Figure 17. The baseline is the average of acoustic heatmaps across the scene, therefore small

directional information is not well preserved. Our proposed model matches the target orientation well, with minor loss of local details.

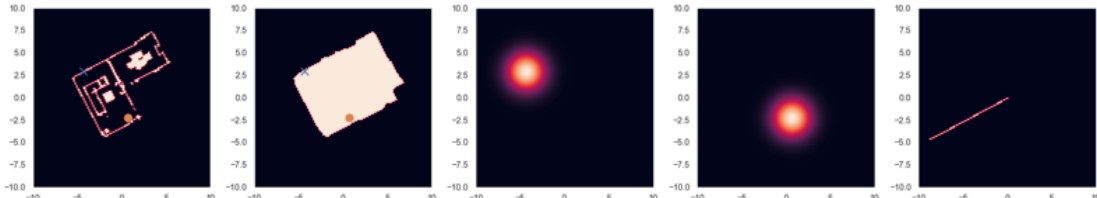

Figure 15: Input features for the spatially-dependent case. An extra channel denotes the scene orientation after rotation is added.

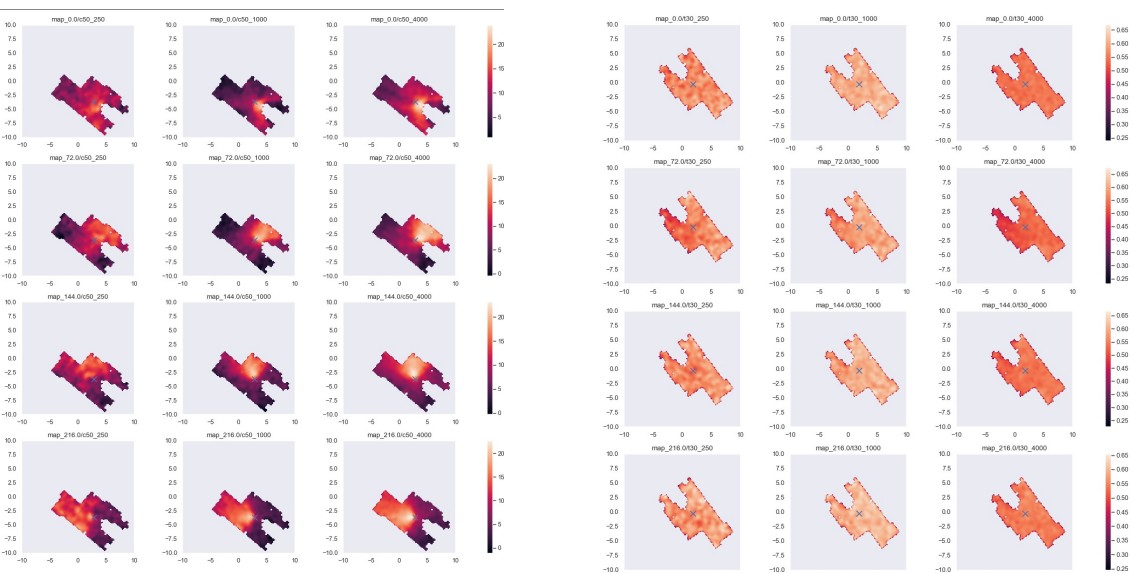

Figure 16: Example of spatially-dependent acoustical parameters. (left) $C_{50}$ computed at 4 fixed orientations and 3 frequency bands show significant differences between front and back, especially at high frequencies. On the other hand, $T_{30}$ (right) shows little variance.

# E ABLATIONS

## E.1 ARCHITECTURE AND FEATURES

In this section, we explore the functions of the model components and the features. Table 4 shows the performance of the model with different configurations for the architecture and input features. For the Replica dataset, there is little difference across most configurations, but there are noticeable cases. First, adding dropout of data augmentation (Basic+drop+aug) shows a marked improvement over the basic model. This is the main configuration that we used for all the main experiments in the paper. Secondly, removing the source (Basic+drop+aug+NoSrc) significantly lowers the performance. This is because the model has no way of knowing where the target source should be located, and energy based parameters heavily depend on this position. On the other hand, removing the reference RIR has little impact. This means that the geometric information is enough to predict the acoustic parameters of the scene, mainly because there is little acoustic diversity in the Replica dataset. Finally, variants that encode the positions for source and receiver differently (Basic+drop+aug+NoSoftSrc, Basic+drop+aug+SrcPosEnc) have limited impact.

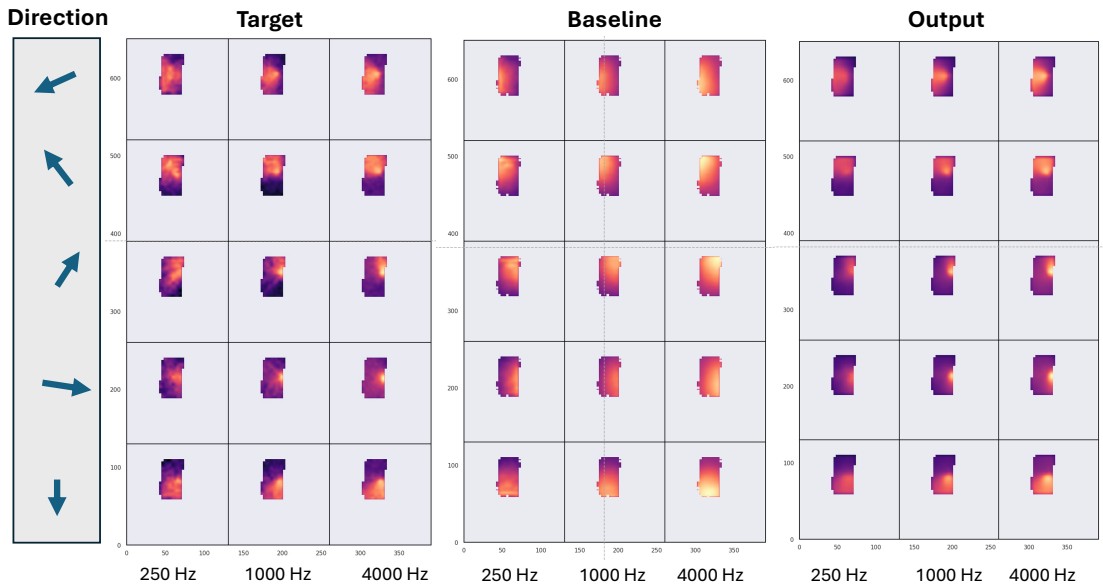

Figure 17: Example of results from the SAPEtask for the spatially-dependent case. Here we compute single acoustic parameter $C_{50}$ at 5 fixed directions in the scene (shown in the far left column), at 3 frequency bands. The SCENE AVG MAP baseline is not able to capture the specific directional patterns and shows little variation across target orientations. The output of the model (with pose) shows the patterns clearly.

For the MRAS dataset, most of the results follow similar trends, with one noticeable exception. Here, removing the reference RIR (Basic+drop+aug+NoRir) is substantially worse. This is because the acoustic diversity of the MRAS dataset is much larger (see Appendix A), and each scene has 5 acoustic configurations, so the model needs the RIR to extract this information.

Table 4: Experimental results for different ablations of the model architecture of the proposed model, when predicting 4 acoustic parameters at 6 frequency bands.

| Model | Dataset | $C_{50}$ (dB) ↓ | $T_{30}$ (%) ↓ | DRR(dB) ↓ | EDT(%) ↓ | loss ↓ | SSIM ↑ |
|---|---|---|---|---|---|---|---|
| Basic | Replica | 1.99± 0.93 | 0.14± 0.06 | 1.55± 0.58 | 20.51±10.55 | 0.12± 0.06 | 0.42± 0.10 |
| Basic+drop | Replica | 1.97± 1.03 | 0.13± 0.07 | 1.52± 0.60 | 19.27±11.23 | 0.12± 0.06 | 0.44± 0.11 |
| (Ours) Basic+drop+aug | Replica | 1.82± 0.85 | 0.11± 0.06 | 1.41± 0.46 | 17.46±10.96 | 0.10± 0.05 | 0.50± 0.08 |
| Basic+drop+aug+NoRir | Replica | 1.82± 0.94 | 0.11± 0.07 | 1.41± 0.46 | 18.28±10.79 | 0.10± 0.06 | 0.48± 0.10 |
| Basic+drop+aug+NoSrc | Replica | 2.60± 0.98 | 0.13± 2.31 | 1.78± 0.65 | 50.75±36.15 | 0.12± 0.07 | 0.46± 0.06 |
| Basic+drop+aug+NoSoftSrc | Replica | 1.75± 0.89 | 0.10± 0.06 | 1.38± 0.46 | 15.69± 8.40 | 0.10± 0.05 | 0.51± 0.08 |
| Basic+drop+aug+SrcPosEnc | Replica | 1.78± 0.85 | 0.10± 0.06 | 1.39± 0.44 | 17.02± 9.79 | 0.10± 0.05 | 0.49± 0.08 |
| Basic+drop+aug+FloormapNoise | Replica | 1.85± 0.87 | 0.11± 0.06 | 1.41± 0.44 | 17.13± 9.55 | 0.10± 0.06 | 0.48± 0.09 |
| Basic+drop+aug+disc | Replica | 1.88± 0.91 | 0.13± 0.07 | 1.45± 0.46 | 19.61±11.37 | 0.11± 0.06 | 0.43± 0.10 |
| Basic+drop+aug+disc+cond | Replica | 1.84± 0.83 | 0.11± 0.05 | 1.42± 0.45 | 16.95± 8.81 | 0.10± 0.05 | 0.48± 0.08 |
| Basic+drop+aug+mdiscs+cond | Replica | 2.07± 0.83 | 0.13± 0.06 | 1.56± 0.49 | 21.50±13.60 | 0.12± 0.06 | 0.40± 0.06 |
| Basic | MRAS | 2.19± 1.07 | 0.19± 0.12 | 1.43± 0.54 | 22.17±14.32 | 0.12± 0.08 | 0.59± 0.09 |
| Basic+drop | MRAS | 2.08± 1.03 | 0.17± 0.12 | 1.37± 0.51 | 20.43±14.07 | 0.11± 0.07 | 0.61± 0.09 |
| (Ours) Basic+drop+aug | MRAS | 1.87± 0.89 | 0.17± 0.10 | 1.34± 0.44 | 19.70±12.27 | 0.10± 0.06 | 0.58± 0.07 |
| Basic+drop+aug+NoRir | MRAS | 2.74± 1.55 | 0.32± 0.21 | 1.67± 0.74 | 30.09±19.47 | 0.16± 0.12 | 0.53± 0.10 |
| Basic+drop+aug+NoSrc | MRAS | 2.52± 1.35 | 0.18± 0.12 | 1.61± 0.62 | 25.12±18.00 | 0.13± 0.08 | 0.55± 0.09 |
| Basic+drop+aug+NoSoftSrc | MRAS | 1.92± 0.91 | 0.17± 0.11 | 1.36± 0.46 | 19.86±12.91 | 0.11± 0.06 | 0.58± 0.07 |
| Basic+drop+aug+SrcPosEnc | MRAS | 1.87± 0.90 | 0.17± 0.11 | 1.34± 0.45 | 19.58±12.90 | 0.10± 0.06 | 0.58± 0.07 |
| Basic+drop+aug+disc | MRAS | 2.06± 1.04 | 0.18± 0.10 | 1.44± 0.51 | 21.89±13.90 | 0.11± 0.07 | 0.56± 0.08 |
| Basic+drop+aug+disc+cond | MRAS | 2.00± 1.02 | 0.19± 0.11 | 1.41± 0.49 | 22.11±14.65 | 0.11± 0.06 | 0.56± 0.08 |
| Basic+drop+aug+mdiscs+cond | MRAS | 2.45± 1.12 | 0.24± 0.15 | 1.63± 0.58 | 26.32±16.08 | 0.13± 0.07 | 0.47± 0.07 |

## E.2 FLOORMAP PROCESSING

Here we analyze the floormap extraction process, and the results are shown in Table 5. The default floormap uses a single slice at the center height of the scene, covering a maximum total area of 100 square meters (denoted in the table as $10 \times 10$). First we test the slicing of the 3D model. In general, slices closer to the floor will include more information about furniture, while slices closer to the ceiling will mostly only contain the walls and dooframes. However, there is little difference between slicing modes. This is mostly likely because late reverberation acoustic parameters are not heavily influenced by small surfaces and fine furniture details. Moreover, adding more slices (5 random slices) also does not improve the performance.

Next, we look into the total map area. In all cases, the floormaps have a fixed resolution of $128 \times 128$ pixels, but the area covered in this map can be varied. The default size is $10 \times 10$ meters, where each pixel covers about 15 centimeters. If the total map area is larger, then each pixel will also cover a larger area, rendering minor spatial details absent in the floormap and acoustic map.

The results show that overall, larger area maps have lower errors. However, this is not because the model is better, but rather because the task is easier. For very large maps, most scenes will have very little variance, so predicting a single mean parameter value over the entire map is often sufficient. Based on this result, we choose the floormap configuration that has the smallest area, while still accommodating for all scenes in both datasets used in the experiments. This allows for comparison across them.

Table 5: Experimental results for different ablations of the input features, when predicting 4 acoustic parameters at 6 frequency bands.

| Model | Dataset | Slice Method | Map Area | $C_{50}$ | $T_{30}$ | DRR | EDT | loss | SSIM $\uparrow$ |
|---|---|---|---|---|---|---|---|---|---|
| Basic | replica | center | 10x10 | 1.88± 0.87 | 0.13± 0.07 | 1.42± 0.44 | 0.18± 0.10 | 0.11± 0.05 | 0.44± 0.10 |
| Basic | replica | random | 10x10 | 1.82± 0.89 | 0.11± 0.07 | 1.42± 0.53 | 0.17± 0.09 | 0.10± 0.06 | 0.50± 0.09 |
| Basic | replica | floor | 10x10 | 1.82± 0.92 | 0.12± 0.08 | 1.44± 0.55 | 0.17± 0.10 | 0.11± 0.06 | 0.49± 0.11 |
| Basic | replica | ceiling | 10x10 | 1.90± 0.92 | 0.12± 0.08 | 1.45± 0.47 | 0.18± 0.11 | 0.11± 0.06 | 0.46± 0.09 |
| Basic | replica | 5×random | 10x10 | 1.83± 0.85 | 0.13± 0.09 | 1.43± 0.49 | 0.17± 0.08 | 0.11± 0.06 | 0.46± 0.11 |
| Basic | replica | 5×random+var | 10x10 | 1.80± 0.87 | 0.15± 0.09 | 1.41± 0.49 | 0.18± 0.09 | 0.11± 0.05 | 0.43± 0.12 |
| Basic | replica | center | 15x15 | 1.82± 1.00 | 0.13± 0.09 | 1.28± 0.55 | 0.18± 0.11 | 0.10± 0.06 | 0.51± 0.13 |
| Basic | replica | center | 20x20 | 2.05± 0.94 | 0.10± 0.05 | 1.34± 0.56 | 0.19± 0.11 | 0.11± 0.06 | 0.50± 0.06 |
| Basic | replica | center | 30x30 | 1.85± 1.15 | 0.11± 0.07 | 1.16± 0.62 | 0.17± 0.10 | 0.10± 0.06 | 0.54± 0.13 |
| Basic | replica | center | 50x50 | 1.65± 0.96 | 0.10± 0.06 | 1.08± 0.59 | 0.15± 0.09 | 0.09± 0.05 | 0.56± 0.12 |

## F RENDERING PLAUSIBLE RIRS FROM ACOUSTIC PARAMETERS

The spatially-distributed acoustic parameters predicted in this work provide perceptually meaningful information about how the acoustics change across a scene. These parameters can also be used to condition algorithmic reverberators to synthesize late reverberation, as shown in Equation 3. A reverberator could take the form of filtered noise with decay (Välimäki et al., 2012), Feedback Delay Networks (Schlecht & Habets, 2019), or a fully neural network-based approach (Lee et al., 2022), among others. The design of the artificial reverberator and the specific conditioning method are out of scope of this work. However, to demonstrate this full pipeline, here we present a simple proof of concept of how acoustic parameters can be used to synthesize RIRs.

We use a commercially available tool, META XR Audio SDK (Audio SDK), an algorithmic reverberator, to generate RIRs. This engine takes a set of reverberation times (T60) at multiple frequency bands as inputs, and uses an artificial reverberator to render RIRs by matching the target T60s. We informally compare the sound quality between a reference RIR, an RIR generated by conditioning on the output of the model, and an RIR generated by conditioning on the noisy outputs of the baseline method. The RIRs conditioned on the model predictions are closer to the reference than those conditioned on the baseline, as shown by the spectrograms in Figure 18. The STFT of the RIR conditioned with the baseline shows some noticeable differences, where the high frequencies decay faster, low mid frequencies decay longer.

It should be noted that here we do not consider the early reflections; this follows other approaches (Mezza et al., 2024) in virtual acoustics that focus only on the late reverberation part.

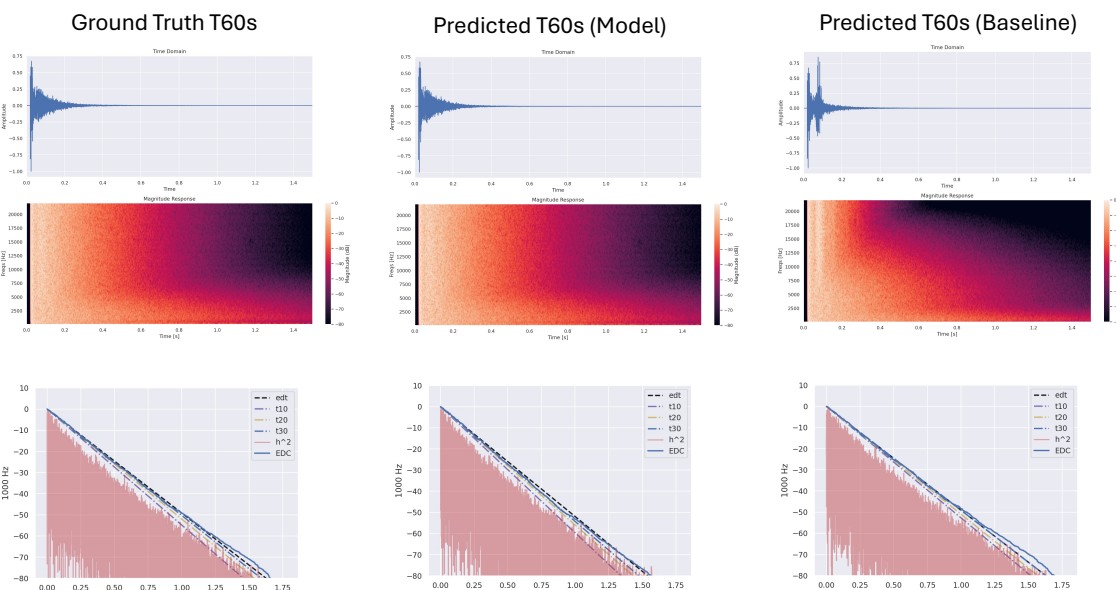

Figure 18: Proof of concept of RIRs generated by conditioning an algorithmic reverberator with acoustic parameters. We show (top row) the time domain RIR, (middle row) the STFT, and (bottom row) the computed slope for reverberation time. (left) a RIR is generated with filtered exponential noise to match the ground truth T60 values. (middle) The RIR is generated with the T60 predicted by our proposed approach. (right) The RIR is generated with the T60s predict by the baseline.

