# OpenReview forum: "Novel View Acoustic Parameter Estimation"
_ICLR.cc/2025/Conference — Submitted to ICLR 2025_

### Official Review · Reviewer_uQPs · 2024-11-01

**Soundness:** 2
**Presentation:** 2
**Contribution:** 2
**Rating:** 3
**Confidence:** 4

**Summary:**

This paper proposed a novel novel view acoustic parameter estimation framework by learning a floor plan like acoustic heatmap. Instead of requiring to know full 3D geometry, this framework uses a 2D floormap and room impulse response (RIR) for inference, translating them into heatmaps representing acoustic parameters. Apart from this task, a new dataset with diverse apartment layouts was used to demonstrate that the proposed model outperforms baselines, offering efficient parameter predictions for varied source-receiver positions in complex, multi-room scenes​. The new dataset is built from SoundSpaces 1.0 simulator, requiring highly dense RIR sampling.

**Strengths:**

1. This paper discusses a challenging yet important task -- estimating acoustic parameters.

2. The introduced large-scale new dataset for future exploration (although I have concerns regarding this dataset)

**Weaknesses:**

1. **Motivation**: While it seems natural to use an image-to-image conversion to learn acoustic heatmap from 2D floorplan map, the motivation of using 2D floorplan as input to learn acoustic parameter isn't convencing. First, 2D floorplan input used in this paper is a degragated representation of the 3D room scene. Furthermore, this paper uses small-resolution (128x128) 2D floorplan as input without explicit room constructional detail. Essentially, the acoustic heatmap's goal is to predict RIR for arbitrary source and receiver position, I didn't the see the need to format it as floorplan like map. Second, obtaining 2D floorplan map is a challenging task in real-scenario, which makes the argument "learning from floorplan map" less practical in real world where the 3D mesh of the a room scene is usually missing. What is the advantage of using 2D floorplan to learn acoustic parameters?

2. **Acoustic Parameter Definition Unclear**: throughout this paper, it claims the task is to estimate the acoustic parameter for novel view, but in essense it estimates RIR. Equalizing RIR with acoustic parameter is confusing, as acoustic parameter usually indicates all relevant characteristics for a room scene that govern sound propagation behaviour, such as room layout, construction material and furniture. RIR is the overall resultant of all the acoustic parameters. To me, this work estimates the RIR, not directly the acoustic parameter. What is the concrete definition of acoustic parameter in the context of this paper? How is it related to RIR?

3. **Acoustic Heatmap**: Based on acoustic heatmap visualization in this paper, it looks like loudness map that depends on the source emitter position (Figure 1 and Figure 3 in this paper). How are all acoustic parameters encoded in the heatmap needs more clarification. The acoustic parameters should be independent on if there is any source eimitter. If so, how to understand the acoustic parameter via heatmap perspective? How does each channel of feature in the learned heatmap related to acoustic parameters?

**Questions:**

1. Read the weakness section.

2. L220-222: These parameters are computed from RIRs captured at a discrete set of receiver locations. **What are these parameter?**

3. Equation 4, 5: I read this paper a couple of times but still confused, the input is the stack of 5 maps (Fig. 3), what is the corresponding ground truth heatmap? **How to obtain the acoustic heatmap during training?** This echoes my confusion of the definition of acoustic parameters heatmap.

4. **MRAS dataset**: L320-323, it writes: it includes a large collection of scene geometries. What collections? It is the same dataset of SoundSpace 1.0 introduced dense RIRs on Replica or Matterport 3D room scene?

5. **Experiment**: this paper writes in L62, previous work is limited by poor generalization to new scenes. But the experiment didn't explicitly discuss the generalization capability of their framework. Moreover, experiment on real-world data is missing, which makes the framework less convencing.

6. **Missing relevant work discussion**, e.g. [1] Y. He, et al., Deep Neural Room Acoustics Primitive, ICML24. [2] Das, O. et al., Room impulse response interpolation from a sparse set of measurements using a modal architecture. ICASSP 2021.

---

> ### Author Response · Authors · 2024-11-27
>
> We thank the reviewer for their critical assessment of our work. In the following we address their concerns point by point. Corresponding changes to the manuscript are highlighted in blue.
>
> ### Weaknesses ###
> - **W1.** While we do use 3D meshes to obtain the floormaps in this work, this is not a requirement more generally.  Floormaps can be obtained easily for real-world rooms from building construction plans, or by roughly measuring the area manually.  There is a significant amount of work on floormap extraction from other low-cost modalities that do not require complete geometric knowledge of the scene, such as from sparse images [1] or from IMU data associated with walking trajectories [2]; however, we assume that demonstrating these methods are outside the scope of this work.  We utilize a 3D mesh-to-floormap extraction process only as an illustrative demonstration, and because we are using synthetic room data for training and testing.  We add additional text in Section 3.2 to clarify this.
>
>     We choose to use a floormap as it is a simplified, low-complexity representation of a scene, and demonstrate that we can infer acoustic parameters without the need for more dense information such as natural images, meshes, or material properties. Regarding the small resolution of the floormap, based on our ablation study of slicing heights (Appendix E.4), it appears that fine furniture detail is not required to infer acoustic parameters accurately - the most important information includes room boundaries and doorways.
>
>     Finally, the advantage of predicting a heatmap of acoustic parameters in the shape of the floormap is that, with a single inference step, we are able to obtain the acoustic parameters for the entire scene, without requiring an explicit source-receiver query at the model output stage.  This allows us to model how the sound characteristics -- and in turn, RIRs -- change over the space in the room.
>
> - **W2.** There may be a misunderstanding here. Firstly, our models do not predict RIRs; we predict a 2D map corresponding to the room's floormap, where the pixel values are determined by acoustic parameter values. In room acoustics, ``acoustic parameters" is defined as a set of parameters that describe key characteristics of an RIR [3], and can be used to reconstruct plausible RIRs in conjunction with other information [4].  In this work, we want to predict acoustic parameters since we can use this information to condition an algorithmic reverberator to generate a synthetic, plausible RIR, while using lower complexity visual and acoustic context as input to the model.  We have clarified this further in the Introduction.
>
>
> - **W3.** Acoustic parameters, just like an RIR, depend on both the source and receiver position.  The heatmap that we predict can be thought of as a collection of the acoustic parameter values at all possible receiver locations paired with a given source position.  Therefore, in Eq. 2 of the manuscript, $\mathcal{A}_{E_t}$ depends only on the target source $E_t$.  Each channel of the model output provides such a heatmap for a single acoustic parameter (T60, C50, etc) at a particular frequency band.  For example, in the experiment described in Table 1, four acoustic parameters (C50, T60, DRR, EDT) for each of six frequency bands ( centered at 125, 250, 500, 1000, 2000, 4000 Hz) leads to 24 output heatmaps produced by the model for a given source $E_t$.  This explanation has been added to the text, in Section 3.3.
>
> [1] Arnaud Gueze, Matthieu Ospici, Damien Rohmer, and Marie-Paule Cani. Floor plan recon-struction from sparse views: Combining graph neural network with constrained diffusion. In Proceedings of the IEEE/CVF International Conference on Computer Vision, pages 1583–1592, 2023.
>
> [2] Claudio Mura, Renato Pajarola, Konrad Schindler, and Niloy Mitra. Walk2map: Extracting floor plans from indoor walk trajectories. In Computer Graphics Forum, volume 40, pages 375–388. Wiley Online Library, 2021
>
> [3] Philipp G¨otz, Cagdas Tuna, Andreas Brendel, Andreas Walther, and Emanu¨el AP Habets. Blind acoustic parameter estimation through task-agnostic embeddings using latent approximations. In 2024 18th International Workshop on Acoustic Signal Enhancement (IWAENC), pages 289–293. IEEE, 2024
>
> [4] Alessandro Ilic Mezza, Riccardo Giampiccolo, Enzo De Sena, and Alberto Bernardini. Data-driven room acoustic modeling via differentiable feedback delay networks with learnable delay lines. EURASIP J. Audio Speech Music. Process., 2024(1):51, 2024

---

> ### Author Response · Authors · 2024-11-27
>
> ## Questions
> - **Q1.** We have addressed the weaknesses directly in our response above this one.
>
> - **Q2.** 	We use standard parameters (for example, also studied in acoustics [3]) and described in ISO standards [5]. These parameters can be computed as broadband quantities, or at specific frequency bands, by first filtering the RIR and then computing the parameters on the filtered RIRs. Specifically, for the experiments in Table 1, the parameters we study are C50, T60, EDT, and DRR. They are computed at six frequency bands, centered at 125, 250, 500, 1000, 2000, 4000 Hz. This gives us a total of 24 output channels/heatmaps predicted by our models. In Table 2, we study the case of spatially-dependent parameter prediction, where the acoustic parameters are computed from spatially filtered signals. We use C50 at three frequency bands (centered at 250, 1000, 4000Hz), computed for five fixed orientations, for a total of 15 channels.
>
> - **Q3.** Following the clarification in **Q2** on the acoustic parameters used. We obtain the ground truth heatmaps by processing the acoustic parameters computed on the ground truth RIRs. This process uses masked average pooling to assign the value of the parameters to nearby pixels. The mask ensures that only pixels with a receiver contribute. This is needed because the number of pixels does not necessarily match the number of receiver positions. We also apply a Gaussian blur to the ground truth heatmaps as a low pass filters that attenuates large variations in neighboring pixels. This is because acoustic parameters typically do not have very large variations in close proximities. We have edited section 3.2 of the manuscript, to improve clarity where this process is described.
>
> - **Q4.** The scene geometries in the MRAS dataset are not from any existing dataset such as SoundSpaces or Matterport, but custom-generated to provide a larger dataset with more complex geometries and material properties.  Some examples of the floormaps of the MRAS scenes are shown in Figure 1, and a more detailed look at the statistics of the dataset are shown in Appendix A; they suggest that MRAS has greater geometric and acoustic diversity than existing datasets like SoundSpaces 1.0 or Matterport 3D.  We intend to release the MRAS dataset for public use, contingent on paper acceptance.
>
>
> - **Q5.** By generalization, we mean that we can infer acoustic parameter for scenes that are not used during training and are only seen at test time.  This is described in Section 5.4 and Table 1.  We concur with the reviewers that training and/or testing on real-world data would further demonstrate generalizability; however, large-scale public datasets are not available for this purpose.  Datasets like [6] or [7] provide a very limited number of scenes and RIRs.
>
>
> - **Q6.** We thank the review for pointing this references. We have incorporated these suggestions into Section 2 of the manuscript.
>
>
> [3] Philipp G¨otz, Cagdas Tuna, Andreas Brendel, Andreas Walther, and Emanu¨el AP Habets. Blind acoustic parameter estimation through task-agnostic embeddings using latent approximations. In 2024 18th International Workshop on Acoustic Signal Enhancement (IWAENC), pages 289–293. IEEE, 2024
>
> [5] ISO 3382-1:2009 Acoustics — Measurement of room acoustic parameters. https://www.iso.org/standard/40979.html, 2009. Accessed: August 2024.
>
> [6] Ziyang Chen, Israel D Gebru, Christian Richardt, Anurag Kumar, William Laney, Andrew Owens, and Alexander Richard. Real acoustic fields: An audio-visual room acoustics dataset and benchmark. In Proceedings of the IEEE/CVF Conference on Computer Vision and Pattern Recognition, pages 21886–21896, 2024
>
> [7] Florian Klein and Sebasti`a V Amengual Gar´ı. The r3vival dataset: Repository of room responses and 360 videos of a variable acoustics lab. In ICASSP 2023-2023 IEEE International Conference on Acoustics, Speech and Signal Processing (ICASSP), pages 1–5. IEEE, 2023

---

### Official Review · Reviewer_tNWT · 2024-11-02

**Soundness:** 3
**Presentation:** 3
**Contribution:** 3
**Rating:** 6
**Confidence:** 4

**Summary:**

The paper introduces a new task, Novel View Acoustic Parameter Estimation (NVAPE), which can be then used to condition a simple reverberator for arbitrary source and receiver positions. The motivation of bringing this task out is to address existed challenges such as poor generalization to new scenes, poor handling of complex geometries, ignoring of directional dependencies, and strict constraint on the input for the previous acoustic modeling methods. As solution, the paper proposed an image-to-image translation style framework to translate floormaps into heatmaps of acoustic parameters. They claim the proposed method can outperform statistical baselines and work for directionally-dependent parameter prediction, requiring only a broad outline of the scene and a single RIR at inference time. A new, large-scale dataset of 1000 scenes consisting of complex, multi-room apartment conditions is also proposed by this paper as the benchmark.

**Strengths:**

+ This paper proposes an interesting task that estimating acoustic parameters given the layout and one reference impulse response. The paper proofs it's feasible to use an image-to-image way to correlate the floormap with the acoustic heatmap and is able to generalize over new rooms as well. It's insightful and potentially can help with other acoustic-visual tasks.

+ The paper introduces a challenging dataset of 1000 scenes consisting of complex, multi-room apartment conditions. The dataset can be good resource for the acoustic learning community.

+ Experiments and visualization analysis show the effectiveness of the proposed method. Details of dataset creating and method implementation are clearly described.

**Weaknesses:**

- Problem Scope. Previous acoustic modeling methods usually output the impulse response wavform, so they can convolve with source sound with the impulse response to render virtual sound that matches the target environment and view pose. However, how to bridge the predicted acoustic parameter with the sound rendering is not explained in this paper. If not targeting on the sound rendering, regardless of presenting this as main application in the introduction, what can we use the parameters for? If sound rendering is indeed the final target, it will be better to include some results, conditioning on the acoustic parameter to generate IR and even target sound, comparing with baselines.

- Fair Comparison Concerns: In the Table 2, the paper compares the proposed method with baselines INRAS and NAF. Though it shows largely accuracy boost, the fairness seems to be an issue. It mentions the acoustic parameters are obtained from predicted IR of INRAS and NAF. However, for fair comparison, we can also compare with these methods by adapting their output from IR wavform to acoustic parameters as well (aligned with the proposed method), at least for INRAS and other baselines such as FewshotRIR [1] that support multi-scene learning. After supervised training and then compare will be more fair. Instead of using an image2image way conditioned on layout and reference IR, these methods rely on modeling the relationship between the target location and references (bounce points or reference locations). By comparing like this, we can better show the advantageous of the proposed method on learning the acoustic parameters.

[1] Few-Shot Audio-Visual Learning of Environment Acoustics. Majumder, S et al.

**Questions:**

Please check the weaknesses section for most of the concerns.
For the task, shall we call it "novel view" acoustic parameter estimation? Since it's only location based, and not condition on any target orientation.

---

> ### Author Response · Authors · 2024-11-27
>
> We thank the reviewer for their critical assessment of our work. In the following we address their concerns point by point. Corresponding changes to the manuscript are highlighted in blue.
>
>
> ### Weaknesses
> - **W1.** Our ultimate goal is indeed rendering of plausible acoustics for virtual and mixed reality applications. For this, as noted by the reviewer, we need RIRs that are perceptually plausible, but not necessarily physically accurate. The task of predicting physically accurate time-domain RIRs from limited information about a scene is challenging. Most other work in deep learning acoustics focuses on predicting RIRs with an emphasis on physical accuracy rather than perception. But this comes at the cost of requiring more complex input information (e.g. full meshes with material properties, or a large number of reference RIRs), or limited generalization to unseen scenes. An alternative approach is to use algorithmic reverberators that are capable of generating synthetic, perceptually plausible RIRs, when conditioned with accurate acoustic parameter values.  Therefore, we predict acoustic parameters needed to drive these reverberators, instead of predicting the RIRs directly.
>
>     To demonstrate this full pipeline, we have modified the manuscript to include an example of this process. We used META XR Audio SDK [1], an algorithmic reverberator, to generate RIRs. This engine uses an artificial reverberator that renders RIRs by matching the target T60s. We informally compare the sound quality between a reference RIR, an RIR generated by conditioning on the output of the model, and an RIR generated by conditioning on the noisy outputs of the baseline. The RIRs conditioned on the model predictions are perceptually close to the reference. Here we do not consider the early reflections; this is similar to other approaches in virtual acoustics that focus only on the late reverberation part, as in [2].  We added this explanation to a new section, Appendix F, in the manuscript.
>
>
> - **W2.** We acknowledge the reviewer's suggestion, but believe our original comparison to NAF and INRAS does make sense.  We extract acoustic parameters from the IRs predicted by these models, since predicting RIRs directly is the approach they take towards accurate sound rendering. We then evaluate the performance of the acoustic parameters against our approach, as RIRs *must* encode acoustic parameters accurately in order to be perceptually plausible for sound rendering.
>
>
> ### Questions
> - **Q1.** We agree with the reviewer's suggestion regarding the title.  While we do use listener pose as an additional input in the spatially dependent parameter case, we have updated the title and corresponding abbreviations for greater clarity.
>
>
>
> [1] Acoustic Ray Tracing for Unity Overview. https://developers.meta.com/horizon/documentation/unity/meta-xr-acoustic-ray-tracing-unity-overview, 2024. Accessed: 22 November 2024
>
> [2] Alessandro Ilic Mezza, Riccardo Giampiccolo, Enzo De Sena, and Alberto Bernardini. Data-driven room acoustic modeling via differentiable feedback delay networks with learnable delay lines. EURASIP J. Audio Speech Music. Process., 2024(1):51, 2024

---

### Official Review · Reviewer_y9ru · 2024-11-02

**Soundness:** 2
**Presentation:** 3
**Contribution:** 2
**Rating:** 6
**Confidence:** 4

**Summary:**

The paper explores estimating room impulse response (RIR) given a spatially distributed pair of sources and receivers' positions.
In particular, the paper proposes a novel way of predicting acoustic maps in the formulation of image-to-image translation, given floor maps and acoustic parameters as input. The proposed method has been evaluated in two datasets, namely SoundSpace and MRAS, constructed to reflect realistic multi-room scenes of typical indoor apartments. Five existing methods have been compared to the proposed method.

**Strengths:**

I find the formulation of RIR estimation into image-to-image translation interesting and novel. The paper is well structured and clearly addresses the problem and limitations of the existing approach. The evaluation result is convincing, and the proposed dataset also seems to reflect a more realistic environment of common apartments than the existing dataset, SoundSpace.

**Weaknesses:**

The authors claim that the proposed method predicts acoustic parameters of 'unseen scenes'. Although I understand the context, it sounds like an overclaim. The proposed model takes a floor map as a geometric input, which requires a full 3D mesh of the scene. Given that the floor map, which requires significant work to build, is already given as input, I feel that it is not an entirely unseen scene. It would make the purpose of the paper clearer for readers by correctly addressing what the proposed aims to do in this regard. I would like to hear what the authors think about this.

**Questions:**

(1) Regarding the weakness that I mentioned, how costly is it to extract the full mesh of the scene to generate the floor map?

(2) The author mentions that the floor map is constructed by slicing at certain heights. How were the certain heights modelled? What is the impact of the different slicing heights?

(3) What are the parameters of low-pass filters?

---

> ### Author Response · Authors · 2024-11-27
>
> We thank the reviewers for their critical assessment of our work. In the following we address their concerns point by point. Corresponding changes to the manuscript are highlighted in blue.
>
> ### Questions
> - **Q1.** While we do use 3D meshes to obtain the floormaps in this work, this is not a requirement more generally.  Floormaps can be obtained easily for real-world rooms from building construction plans, or by roughly measuring the area manually.  There is a significant amount of work on floormap extraction from other low-cost modalities that do not require complete geometric knowledge of the scene, such as from sparse images [1] or from IMU data associated with walking trajectories [2]; however, we assume that demonstrating these methods are outside the scope of this work.  We utilize a 3D mesh-to-floormap extraction process only as an illustrative demonstration, and because we are using synthetic room data for training and testing.  We add additional text in Section 3.2 to clarify this.
>
>     Finally, we claim that the work generalizes to unseen scenes, because we test on room scenes that have not been used in training. Most state-of-the-art work based on implicit representations such as NAF or INRAS are limited to interpolation within a single scene, and cannot generalize to completely different geometries.
>
>
>
> - **Q2.** 	We have conducted an ablation study on floormap construction, detailed in appendix E.4. In general, we found little variation in performance based on how maps are constructed. This is mainly because the acoustic parameters we tested do not depend much on fine details in the scene like furniture. The most important features are walls that define the scene boundaries and doorways that connect distinct rooms. It is possible that the slicing height may impact the model's performance in a heavily cluttered scene, which in turn may impact the distribution of acoustic parameters; however, these types of scenes are unlikely in synthetic datasets and real-world scenes.
>
>     We have highlighted the text clarifying this in Appendix E.4.
>
>
>
>
> - **Q3** 	The low pass filter is a 2D gaussian with a kernel size of 9x9 and standard deviation of 1.  We have added this detail to Section 3.2.
>
>
>
>
> [1] Arnaud Gueze, Matthieu Ospici, Damien Rohmer, and Marie-Paule Cani. Floor plan recon-struction from sparse views: Combining graph neural network with constrained diffusion. In Proceedings of the IEEE/CVF International Conference on Computer Vision, pages 1583–1592, 2023.
>
> [2] Claudio Mura, Renato Pajarola, Konrad Schindler, and Niloy Mitra. Walk2map: Extracting floor plans from indoor walk trajectories. In Computer Graphics Forum, volume 40, pages 375–388. Wiley Online Library, 2021

---

> > ### Comment · Reviewer_y9ru · 2024-12-02
> >
> > I appreciate the authors' responses to my concerns. I believe the revised version will make the scope of the paper clearer.

---

### Meta-Review · Area_Chair_dZH5 · 2024-12-18

**Metareview:**

This paper investigates the problem of novel view acoustic parameter estimation and proposes a method for novel view acoustic synthesis, that is, an intermediate representation from which Room Impulse Responses (RIRs) for unseen source and receiver positions in a scene can be generated. 2D spatially distributed acoustic parameters for an unseen scene are estimated casting the problem as an image-to-image translation task, just using in input a simple 2D floormap and a reference RIR. A new, large-scale dataset of 1000 scenes consisting of complex, multi-room apartment conditions (different source-receiver positions) is also proposed as benchmark.

This paper received contrasting reviews: a clear negative (3) and two marginally positive (6, 6) ratings are the scores originally assigned and also maintained after the authors' rebuttal.

Positive points relate to the original formulation of the problem, good explanation of the methodology including its limitations, and convincing experimental evaluation, including the proposal of the new dataset.

Negative points address the unclear actual scope of the problem, also related to the experimental analysis, namely, how to ultimately prove the goodness of the method. Other issues still regard the experimental analysis, specifically, about the unfairness of the comparative tests and the visualization of the acoustic heatmaps. In a similar vein, it was also argued about the acoustic parameter specification and estimation re RIR generation, that is, how much these parameters are suitable to lead to a correct estimation of the RIR.

The authors' rebuttal replied to all main comments, but did not move the original ratings: it addresses satisfactorily some points but leaves other unclear.

AC has read the reviews, the rebuttal and the discussion and, despite the majority of slightly positive ratings, he is more in line with the most critical reviewer.
In particular, the question about the relation between RIR and acoustic parameters has been tackled by inserting an experiment in Appendix F, but the final performance was just "informally" evaluated, and this is not sufficient. In general, the actual scope of the work in relation to resulting virtual sound rendering (as the result of the convolution of the impulse response and source sound) in novel views, does not seem well addressed by the authors' discussions. The raised issues regarding the use of solely a 2D map and a reference RIR (much less info than former methods), which is also quoted as one of the main points of the works, has been discussed and might also be reasonable as long as one can demonstrate that the final goal is reached (virtual audio rendering), but this does not seem to be sufficiently considered. Yet, it results unclear how that a 2D map (and a reference RIR) used for parameter estimation can consider the complexity of an environment, e.g., the room ceiling, which is actually an important part to consider in the acoustic propagation.
Also the fact that some important acoustic parameters are not considered in the estimation is not helping the overall narrative at the basis of this work.

For these reasons, this paper cannot be recommended for acceptance at ICLR 2025.

**Additional Comments On Reviewer Discussion:**

See above

---

### Decision · Program_Chairs · 2025-01-22

Reject